# Single-molecule FRET uncovers hidden conformations and dynamics of human Argonaute 2

Sarah Willkomm [1✉], Leonhard Jakob[1,4], Kevin Kramm[1], Veronika Graus[2], Julia Neumeier[2], Gunter Meister [2] & Dina Grohmann [1,3✉]

Human Argonaute 2 (hAgo2) constitutes the functional core of the RNA interference pathway. Guide RNAs direct hAgo2 to target mRNAs, which ultimately leads to hAgo2-mediated mRNA degradation or translational inhibition. Here, we combine site-specifically labeled hAgo2 with time-resolved single-molecule FRET measurements to monitor conformational states and dynamics of hAgo2 and hAgo2-RNA complexes in solution that remained elusive so far. We observe dynamic anchoring and release of the guide's 3'-end from the PAZ domain during the stepwise target loading process even with a fully complementary target. We find differences in structure and dynamic behavior between partially and fully paired canonical hAgo2-guide/target complexes and the miRNA processing complex formed by hAgo2 and pre-miRNA451. Furthermore, we detect a hitherto unknown conformation of hAgo2-guide/target complexes that poises them for target-directed miRNA degradation. Taken together, our results show how the conformational flexibility of hAgo2-RNA complexes determines function and the fate of the ribonucleoprotein particle.

[1] Institute of Microbiology & Archaea Centre, Single-Molecule Biochemistry Lab, University of Regensburg, 93053 Regensburg, Germany. [2] Laboratory for RNA Biology, University of Regensburg, 93053 Regensburg, Germany. [3] Regensburg Center of Biochemistry (RCB), University of Regensburg, 93053 Regensburg, Germany. [4] Present address: Department of Pharmacology and Toxicology, Institute of Pharmacy, University of Regensburg, 93053 Regensburg, Germany. ✉email: sarah.willkomm@ur.de; dina.grohmann@ur.de

RNA interference (RNAi) is a process to regulate gene expression at the posttranscriptional level[1]. It is based on RNA-induced silencing complexes (RISCs) composed of Argonaute (Ago)[2] and RNA guide and target strands. Guide strands originate from microRNA (miRNA) or small interfering RNA (siRNA) duplexes. After removal of the passenger strand from the duplex, the remaining guide RNA directs Ago to complementary target RNAs. SiRNAs display extensive complementarity to their target and induce human Ago2 (hAgo2)-mediated target RNA cleavage[3,4]. MiRNAs are only partially complementary to mRNA targets and translation is inhibited followed by mRNA decay via cellular degradation pathways[5].

Human Ago2 and its full-length prokaryotic counterparts exhibit a conserved bi-lobal organization[6–13]. The N-terminal lobe encompasses the N-terminal and the PAZ domain and the C-terminal lobe consists of the Mid and the catalytic PIWI domain, which adopts an RNase H-like fold and harbors the catalytic site[3]. The 5'-end of the guide strand is anchored in the Mid domain, whereas the 3'-end is attached to the PAZ domain[6–8,11,13–16]. The N-terminal domain assists guide loading and target cleavage[17,18] and is thought to fulfill a 'wedging' function important for removal of passenger strands and processed substrates[19].

Guide strands bound to Ago are divided into different functional sections[20] (Fig. S1A). The Mid-bound 5'-nucleotide is called anchor[20] and does not contribute to target binding. The following seven nucleotides (g2 - g8) constitute the seed region[21], which is pre-organized for optimal recognition of complementary target RNAs[6–8,22,23]. Target RNA cleavage by hAgo2 requires base pairing in the seed, the central (g9 – g12) and the supplementary region (g13 – g16)[20,24,25]. However, central base pairing occurs only transiently, even if the target RNA has full base pairing potential with the guide[9]. Notably, in contrast to prokaryotic Agos, base pairing does not extend linearly along the guide in case of hAgo2[10,26]. After seed pairing, supplementary pairing follows, which initiates additional central and/or tail (3'-region) base pairing[9,10,27] and is accompanied by structural rearrangements and relocations of the nucleic acids[8,9]. Eventually, the 3'-end of the guide is released from the PAZ domain[10]. Seed-only pairing is sufficient for cleavage-independent hAgo2-mediated regulation[28], although base pairing beyond the seed enhances in vivo gene regulation and binding specificity[29]. Central pairing is a pre-requisite for cleavage but requires a substantial opening of the central cleft and a movement of the PAZ domain and, hence, causes considerable energy costs[8,9].

Dysregulation of cellular miRNA levels can cause a variety of diseases and consequently, cellular miRNA levels are tightly controlled[30]. Recently, target-directed miRNA degradation (TDMD) was described as one possibility to induce miRNA degradation[10,30–34]. TDMD targets display seed pairing, pairing to the supplementary and the tail region of the guide with a central bulge[10,30,31]. Binding of TDMD targets is accompanied by the release of the guide's 3'-end[10] exposing this end for modification[30,31]. Furthermore, it has been speculated that binding of TDMD targets triggers conformational changes that lead to ubiquitination of hAgo2 resulting in the proteolysis of Ago[33,34].

Here, we succeeded to site-specifically engineer donor and acceptor fluorophores to native hAgo2 allowing us to perform single-molecule FRET (smFRET) measurements. We monitored conformational states of apo hAgo2 and RNA-bound states reflecting the progression of hAgo2 in its catalytic cycle. Notably, we observed conformational transitions in a time-resolved manner providing for example insights into the conformational heterogeneity and flexibility of unliganded hAgo2. We furthermore show that seed pairing between guide and target is sufficient to trigger 3'-end release of the guide from the PAZ domain. We find that the 3'-end can dynamically re-associate with the PAZ domain even when

the target has full complementarity to the guide suggesting that frequent dissociation/re-association prevents degradation of the guide at the 3'-end to ensure multiple turnover cycles that are documented for hAgo2. We discovered a hitherto unseen transient conformational state of fully base-paired hAgo2-guide/target complexes. Dynamic switching between this state and two other conformations probably corresponds to a change between cleavage active and inactive states. Additionally, we identified a unique conformational state for hAgo2 in complex with a guide and TDMD-inducing targets providing a structural basis for the efficient exposure of guide RNAs for TDMD. Taken together, our data reveal hitherto undetected conformations of hAgo2 that form the basis for regulatory processes like TDMD thereby expanding our understanding of the conformational landscape and dynamics of hAgo2.

## Results

**Experimental design to probe hAgo2 conformation and dynamics using smFRET.** So far, smFRET measurements of human RISC were limited to observe conformational changes of bound nucleic acids[35,36] as site-specific labeling of hAgo2 could not be achieved. We recently introduced the SLAM-FRET strategy[37] that allows the site-specific labeling of human proteins expressed in their native human cellular context. We produced three doubly labeled hAgo2 variants. Two of these mutants were employed to analyze rearrangements of the Mid and the PAZ domain along the respective intra-lobal axis, using the N-terminal residue phenylalanine 23 as a second labeling position (hAgo2^N/PAZ (fluorophores coupled to residue 23 and 291) and hAgo2^N/Mid (residue 23 and 511)). We chose position 23 for labeling because of its central position in between the lobes of hAgo2. Crystal structures of hAgo2 reveal that this position at the N-terminus of hAgo2 is stably packed against the PIWI domain[6,7]. Hence, phenylalanine 23, although nominal part of the N-terminal domain, is associated with the PIWI domain and stably positioned as a pivot point in between the lobes. The third mutant used was chosen to monitor inter-lobal domain movements (hAgo2^PAZ/Mid (residue 291 and 511)) (Fig. 1A and S1B). Single labeled hAgo2^PAZ and hAgo2^Mid variants (Fig. S1B) in complex with fluorescently labeled guide or target strands (Supplementary Table S1) reported on relative changes between hAgo2 domains relative to guide or target RNAs. HAgo2 variants were produced and purified as full-length proteins (Fig. S2). RNA guide and target strands and RNA labeling positions (Supplementary Table S1) were chosen based on previous studies[38]. The immobilization of purified and labeled hAgo2 molecules on the surface for total internal reflection fluorescence (TIRF) microscopy using an antibody directed against the N-terminal domain[39] led to a clear enrichment of hAgo2 molecules (Fig. 1B and S3A–C). Alternatively, and as a control, we immobilized hAgo2 using an antibody directed against the PIWI domain (Fig. S3D, E). The PIWI-directed antibody additionally ensures the purification of full-length hAgo2. We verified that modified hAgo2 variants were still active in guide-mediated RNA cleavage (Fig. S4A). However, cleavage is prevented by our measurement conditions (Fig. S4B).

First, we tested binary hAgo2^PAZ*Dy550-guide RNA^14CCy5 complexes (Fig. 1C, i). These measurements resulted in a single homogeneous FRET population that reflects a single conformational state of the guide nucleic acid within hAgo2. This correlates well with information derived from crystal structures of hAgo2-guide complexes. Crystal structures suggest a stable positioning of the guide within hAgo2 by (i) anchored 3'- and 5'-guide ends and (ii) interactions of hAgo2 with the guide backbone that lead to a pre-organized seed region and a stable positioning of the

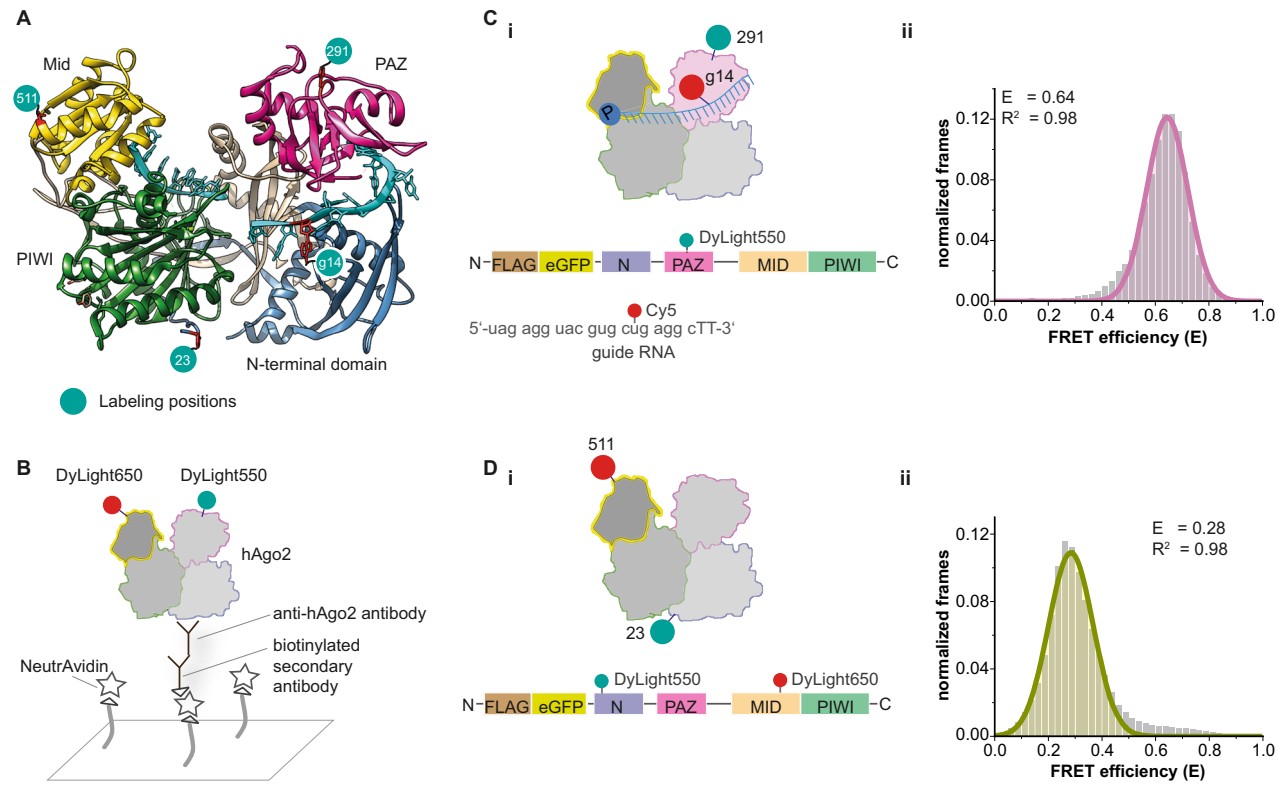

**Fig. 1 Single-molecule FRET measurements using site-specifically labeled human Ago2. A** Crystal structure of a hAgo2-guide RNA complex (pdb 4W5N). Green dots indicate labeling positions. The amino acids (F23, H291, T511) were mutated to p-azido-L-phenylalanine (AzF) to enable site-specific labeling. **B** Immobilization strategy to allow single-molecule FRET measurements using TIRF microscopy. **C** (i) Labeling positions for intermolecular FRET measurements using DyLight™550-labeled hAgo2[PAZ] and a guide RNA[14Cy5]. (ii) FRET efficiency histogram for the setup shown in (i). **D** (i) Labeling positions of hAgo2[N/Mid] for intramolecular FRET measurements. (ii) FRET efficiency histogram for hAgo2[N/Mid]. Both histograms show data of at least three independent measurements. Data were fitted to a single Gaussian equation. Source data provided as a Source Data file.

supplementary region[6–8] (Fig. 1C, ii and S1B). Additionally, we performed smFRET measurements with doubly labeled hAgo2[N/Mid] (Fig. 1D, i) resulting in a single population with a mean FRET efficiency of 0.28 (Fig. 1D, ii). Control measurements with an antibody targeting the PIWI domain (Fig. S3D) confirmed that the antibody does not induce conformational changes as the resulting FRET efficiency of 0.30 is in very good agreement with the FRET efficiency of 0.28 derived from experiments where the immobilization was carried out using the N-terminal antibody (compare Fig. 1D, ii with Fig. S3D). These results demonstrate that information derived from our smFRET measurements match structural information and render this approach suitable to probe the conformational status of hAgo2 throughout its activity cycle.

**Association of the target RNA induces dynamic release of the guide's 3'-end from the PAZ domain.** Structural rearrangements upon target binding occur mainly in the PAZ domain[8,10,26] including the release of the guide's 3'-end from the PAZ domain. This transition is an important step to form catalytically active ternary complexes[9,40–42]. To observe the relative positions of the guide and the PAZ domain upon target loading, we performed intermolecular smFRET measurements using hAgo2[PAZ]-guide RNA[14Cy5] in complex with a range of unlabeled target RNAs to mimic the stepwise formation of the RNA duplex (Fig. 2 and Table 1).

FRET measurements of ternary complexes with a seed-matching target (g2 – g8) resulted in three FRET populations (E = 0.34, 0.47, 0.62; Fig. 2B, i/ii) indicative of three conformational states that differ in the relative position of the PAZ domain to the guide's supplementary region. The mean FRET efficiency of the high FRET population (E = 0.62) is congruent with the

FRET efficiency of the corresponding binary hAgo2-guide complex (E = 0.64) (Fig. 2A, i/ii). We assign the high FRET population to a state in which the 3'-end of the guide is attached to the PAZ domain as observed in the crystal structure of a seed-matched ternary complex (Fig. S5A, B)[8]. The two populations with lower FRET efficiencies (E = 0.47 and E = 0.34) correspond to larger distances between the guide's supplementary region and the PAZ domain. In binary complexes, helix-7 conceals the 3'-part of the seed region but moves in concert with the PAZ domain to open the N-PAZ channel thereby allowing supplementary base pairing[8,9,23]. However, the crystallized state of a hAgo2-guide RNA complex represents a conformation in which the 3'-end of the guide is still anchored in the PAZ domain even in a state in which the seed and supplementary region is paired with the target[9]. This suggests that the release of the guide's 3'-end from the PAZ domain requires extended base pairing beyond the seed and even beyond the supplementary region[8,10,43]. The detection of populations with low FRET efficiencies indicates, however, that the release of the guide's 3'-end occurs at an early stage of guide-target pairing (Figs. 2B and S5B). We conclude that base pairing within the seed region is sufficient to stimulate the release of the 3'-end from the PAZ domain and to initiate conformational changes that are necessary for the formation of catalytically active hAgo2 complexes.

After seed pairing, supplementary base pairing occurs. We mimicked this state using a target RNA with complementarity in the seed and extended supplementary region but with a central bulge (g2 – 8 + g13 – 19). Ago-bound guide-target duplexes with this configuration have been shown to be subject to TDMD[10,30]. Using the biotin-modification at the 5'-end of the target RNA

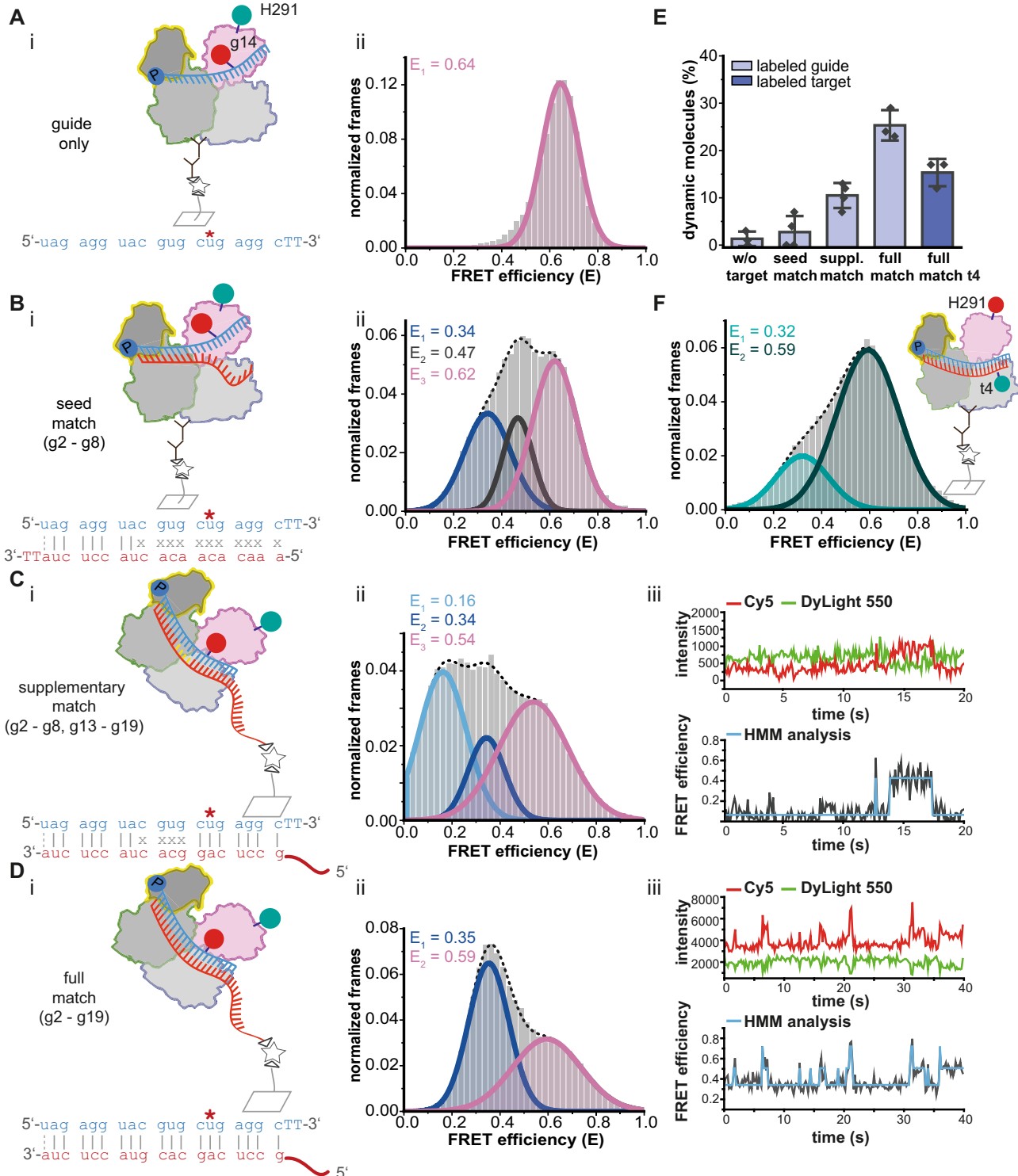

**Fig. 2 Conformational changes accompanying the stepwise target binding process. A–D** (i) Experimental strategies to monitor distance changes between hAgo2[PAZ*DL550] and guide RNA[14Cy5] upon stepwise target RNA binding. Base pairing between g1 and t19 is shown with a dotted line as these bases are unpaired in ternary hAgo2-guide-target complexes. Immobilization of the complexes is achieved via antibodies (**A/B**) or a biotinylated target RNA (**C/D**). Measurements were conducted without target RNA (**A**) and in presence of target RNAs with different degrees of complementarity to the guide RNA (**B**: target RNA[seed], **C**: target RNA[longmm9-12-Biotin], **D**: target RNA[long-Biotin]). (ii) FRET efficiency histograms showing data of at least three independent measurements for the different complexes indicated in (i). **C**, **D** (iii) Representative fluorescence transients with corresponding FRET efficiency trace fitted with Hidden-Markov-Model analysis (HMM) for the complexes depicted in **C**, i and **D**, i. **E** Fraction of dynamic molecules detected in the respective measurements. Bars show the mean of at least three independent measurements and the error bars represent standard deviations (no. of biological replicates: w/o target = 3; seed match = 4; suppl. match = 4; full match = 3; full match t4 = 3). **F** FRET efficiency histogram of three independent measurements of ternary complexes composed of hAgo2[PAZ*DL650], guide RNA and target RNA[4Cy3]. Source data provided as a Source Data file.

instead of an antibody for immobilization ensured that only ternary complexes were observed in smFRET measurements (Fig. 2C, i). We detected three FRET populations (Fig. 2C, ii) (E = 0.16, E = 0.34 and E = 0.54). We propose that the broadly distributed high FRET population is composed of molecules with FRET efficiencies that match the previously observed high FRET (E = 0.62) and medium FRET population (E = 0.47) but can be fitted best with a single population. These populations represent Ago proteins with an attached and released 3'-end of the guide (Fig. S5C). The middle FRET population corresponds to the low FRET population observed with seed-paired ternary complexes (Fig. 2B). The FRET value suggests a large distance between the PAZ domain and the guide supplementary region likely reflecting a hAgo2 state with a released 3'-end of the guide. The same holds true for the low FRET population (E = 0.16). Such a large distance between PAZ domain and the 3'-end of the guide has not been captured by any existing crystal structure. This conformational state is not dependent on the sequence but only dependent on the TDMD characteristics of the miRNA/target RNA pair (Fig. S6A, B). We only observe this low FRET population when using target RNAs with extensive supplementary base pairing and a central bulge (Figs. 2C and S6A; S7A (with a 2 nt central mismatch)). The large distance between the PAZ domain and 3'-end of the guide suggests that the guide is not positioned in the binding channel but relocated in between the N- and the PAZ domain into the direction of the PIWI domain (Fig. S5C). A study demonstrating that kinking occurs in hAgo2-bound guide/target duplexes comprising a TDMD target[10] supports our hypothesis. However, crystal structures of ternary complexes comprising a target with TDMD characteristics do not reveal such a dramatic increase of the distance between the PAZ domain and the guide supplementary region[10] compared to e.g. a seed-matched ternary complex (Fig. S1B). Hence, our measurements indicate a higher degree of kinking. The lack of contacts between hAgo2 and the duplex in the supplementary region[10] enables such a rearrangement. We suggest that pronounced kinking efficiently exposes the 3'-end of the guide rendering it accessible for degrading enzymes or opens up a protein interface for interaction partners to promote ubiquitination of hAgo2[33,34].

In a third step, we performed smFRET measurements of ternary complexes with a fully complementary target RNA (Fig. 2D, i) that is a potential substrate for hAgo2-mediated cleavage, which is, however, prevented by our measurement conditions (Fig. S4B). We observed two distinct populations (E = 0.59 and E = 0.35; Fig. 2D, ii) that match two of the populations found for ternary complexes with a seed-matching target. Thus, the high FRET population most likely represents molecules with the 3'-end of the guide associated with the PAZ domain (Fig. 2A, ii), whereas the low FRET population corresponds to a 3'-end released state accompanied by an opening of hAgo2. Supplementary base pairing is still compatible with 3'-end binding[9]. These data show that a target with full base pairing potential allows for a stable conformation that keeps the guide's 3'-end attached and the central cleft closed. This in turn prevents central base pairing[10]. As suggested by Sheu-Gruttadauria et al.[9], central base pairing - as a pre-requisite for cleavage activity - requires a substantial opening of the central cleft, which causes tension in the 3'-end of the guide forcing the release from the PAZ domain. Sheng et al.[44] analyzed the catalytic pre-cleavage state for *Thermus thermophilus* Ago and showed that the release of the guide's 3'-end from the PAZ domain is among the conformational changes that are a pre-requisite for catalytic activity. Taken together, permanent release of the guide's 3'-end from the PAZ domain is a crucial step towards the formation of catalytically active ternary complexes. Hence, we suggest that the dominant low FRET population in this

measurement (E = 0.35) reflects this catalytically active conformation. We propose that the 3'-end is released and that the guide and target are separated in the supplementary region. In this scenario, the guide protrudes into the N-PIWI channel and the target into the N-PAZ channel as shown in crystal structures of *Thermus thermophilus* Ago[26,44] (Fig. S5D). In addition, biochemical data of eukaryotic Agos confirm that base pairing in the 3'-half of the supplementary region is not required for target RNA cleavage[24,25]. In line with this, the conformation with a mean FRET efficiency of 0.35 is the preferred state in case of fully matched ternary complexes. So far, the structure of an active conformation could not be determined, which led to the suggestion that this conformation is only transiently sampled[9]. In this study, smFRET measurements elucidate the dynamics of a fully matched ternary complex. Approximately 25% of the molecules dynamically switch between a low, intermediate, and high FRET conformation (Fig. 2D, iii; E, S7B–G). In contrast, the fraction of dynamic molecules is lower in case of partially matched targets (Fig. 2E; 2.8% dynamic molecules for seed-only pairing and 10.1% with centrally mismatched targets). Excursion into the intermediate state is a rare event though. Consequently, we only detected this intermediate state in the dynamic molecules of measurements with the different fully matched miRNA/target pairs tested in our study (Fig. 2D,iii, S6C, D and S7F, G). We observed this intermediate state as a stable conformation in case of seed-matched ternary complexes (Fig. 2B, ii). Therefore, we suppose that this population reflects a transition state in between seed plus supplementary pairing with a closed central gate and full pairing with an open central gate[10]. Opening of this gate, that enables for example central pairing, appears to destabilize the intermediate state[9,10]. Although crystal structures show that ternary complexes with full pairing potential are able to adopt a stable conformation without central base pairing[9], cleavage activity and our smFRET data disclose a frequent sampling of other states likely including the catalytically active state. We propose that the energy provided by central and potentially tail pairing destabilizes the closed and especially the intermediate conformation of the central gate. Control experiments (immobilization via an antibody or using targets with different lengths) resulted in highly comparable datasets (compare Figs. 2D and S7B, C) underscoring our findings.

Dwell time analysis of the dynamic molecules revealed that the individual conformational states are populated for 0.5 – 1 s resulting in turnover rates of 1 – 2 s$^{-1}$ (Fig. S7F, ii/G, ii). The rate constants are four to five orders of magnitude faster than the rates determined for building and disruption of base pairs between target RNAs and Ago-bound guide RNAs[20,38]. Hence, we assign the dynamic behavior of ternary complexes to movements of the supplementary guide region or the guide/target duplex, respectively, in combination with rearrangements of the PAZ domain rather than zipping or unzipping of the guide-target duplex.

In the next step, we interrogated complexes that carry the donor dye in the Mid instead of the PAZ domain. Measurement of the binary complex with hAgo2$^{Mid}$ and guide RNA$^{14Cy5}$ resulted in a single FRET population (Fig. S8A). As described above, in this conformation the 3'-end of the guide is attached to the PAZ domain. Addition of a fully complementary target yielded three distinct populations (Fig. S8B and S5F) with a relatively low number of dynamic molecules (Fig. S8C, D) (hAgo2$^{Mid}$: 10% vs. Ago2$^{PAZ}$: 25%). Reduced dynamics can be rationalized by the higher flexibility of the PAZ domain[8,9,23,45]. We assign the high FRET population to centrally mismatched but otherwise fully paired complexes as observed in crystal structures[10] (Fig. S5F). According to our measurements with ternary complexes containing hAgo2$^{PAZ}$, we also expect a conformation, that might represent the catalytically active complex, which supposedly involves a re-

location of the guide's supplementary region into the N-PIWI channel. The low FRET population (Figs. S5F and S8B) likely represents this conformation.

For further analysis of the different guide and target trajectories, we interrogated ternary complexes composed of acceptor-labeled hAgo2$^{PAZ}$, fluorescently labeled target (target RNA$^{4Cy3}$) and unlabeled guide RNA. We observed two broadly distributed populations for these complexes (Fig. 2F). The low FRET population (E = 0.32) likely represents a scenario with base pairing only in the seed region, which results in an increased distance between the PAZ domain and the 5'-region of the target RNA (Fig. S5E). Accordingly, the high FRET population (E = 0.59) represents a conformation with seed and supplementary base pairing. The broad distribution of FRET efficiency values in this population suggest that these conformational states have a certain degree of flexibility. Hence, the high FRET population reflects multiple conformations with the guide's 3'-end attached to the PAZ domain or released and the 5'-region of the target in corresponding states (Fig. S5E). This probably includes the catalytically active conformation (released guide 3'-end, the guide in the N-PIWI channel, the target possibly positioned in the N-PAZ channel near the PAZ domain). This

measurement further corroborates our assumption that guide, and target trajectories are located in the N-PIWI and the N-PAZ channel, respectively. Therefore, our data provide experimental evidence for the suggestion of Kwak and Tomari[19] that the N-terminal domain acts as a wedge to separate the guide and target strand. Using ternary complexes with a labeled guide RNA and a 21 nt unlabeled target RNA, we detected a fraction of 32.3% dynamic molecules. In case of ternary complexes with a 21 nt labeled target RNA, we found that only 15.3% of the molecules are dynamic (Figs. 2E and S7E). This leads to the conclusion that the guide is more flexible than the target upon separation from the target by the N-terminal domain.

**Guide and target binding reduce the conformational flexibility of the N- and C-terminal lobe.** The possibility to perform intramolecular FRET measurements provided the opportunity to access the conformational states of the structurally uncharacterized apo hAgo2. First, we interrogated inter-lobal conformations using hAgo2$^{PAZ/Mid}$. In case of apo hAgo2$^{PAZ/Mid}$ as well as in its guide-bound form we detected a broad distribution of FRET efficiencies, but with three distinguishable populations (Figs. 3A, B and S9A).

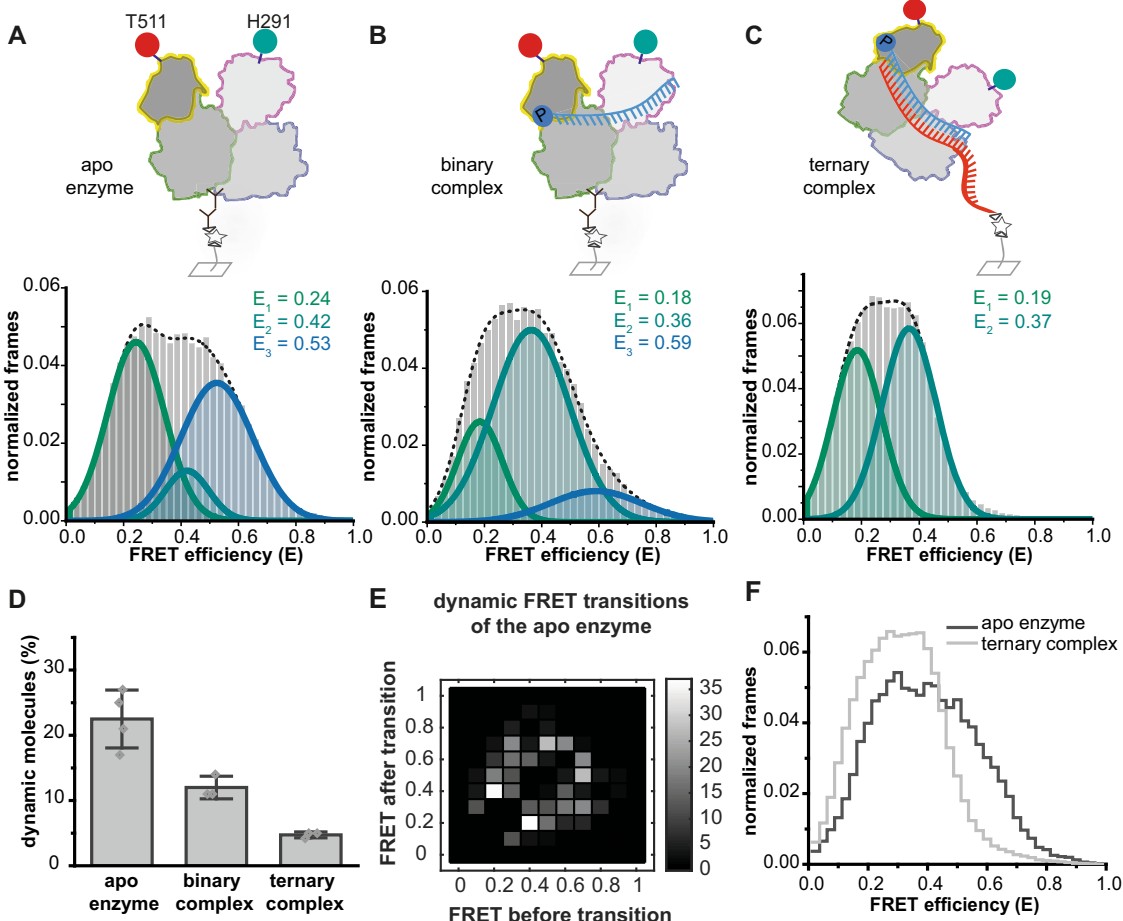

**Fig. 3 Conformational states of the PAZ and Mid domain in the human Ago2 apo enzyme, binary and ternary complex.** Doubly labeled apo hAgo2$^{PAZ/Mid}$ (**A**) and binary hAgo2$^{PAZ/Mid}$-RNA guide complexes (**B**) were immobilized via an hAgo2-directed antibody, whereas ternary hAgo2$^{PAZ/Mid}$-guide-target complexes (**C**) were immobilized via the target RNA$^{long-Biotin}$. FRET efficiency histograms show data from at least three independent measurements. All data were fitted using a Gaussian equation. **D** Fraction of molecules that show dynamics. Bars show the mean of at least three different measurements and error bars indicate the standard deviations (no. of biological replicates: apo enzyme = 4; binary complex = 3; ternary complex = 3). **E** Analysis of all dynamic molecules detected in the measurements presented in (**A**). Transitions between FRET states are presented in a transition density plot. **F** Comparison of FRET efficiency distributions of the apo hAgo2$^{PAZ/Mid}$ enzyme and ternary complexes composed of hAgo2$^{PAZ/Mid}$, guide and target RNA. Source data provided as a Source Data file.

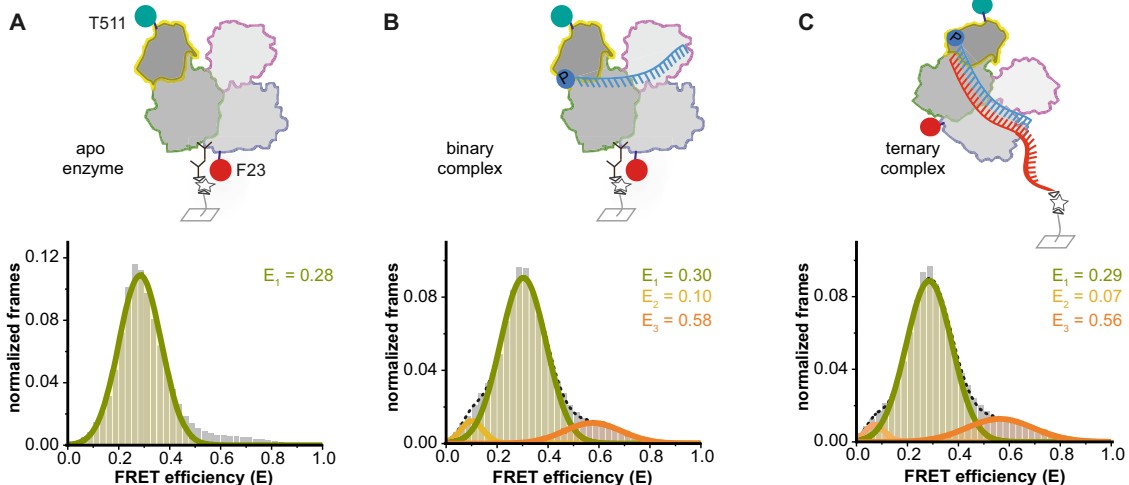

**Fig. 4 Conformational state of the C-terminal lobe of human Ago2 upon binding of guide and target RNA.** Doubly labeled apo hAgo2$^{N/Mid}$ (**A**) and binary hAgo2$^{N/Mid}$-guide RNA complexes (**B**) were immobilized via an hAgo2-directed antibody, whereas ternary hAgo2$^{N/Mid}$-guide-target complexes (**C**) were immobilized via the target RNA$^{long-Biotin}$. All FRET efficiency histograms show data of at least three independent measurements. All data were fitted using a Gaussian equation. Source data provided as a Source Data file.

The purification of hAgo2 expressed in a human cell line carries the risk of purifying hAgo2 molecules that are already bound to a guide strand and/or RNA duplex[4] (Fig. S9A–C). To determine the fraction of hAgo2 molecules, that are loaded with cellular RNAs, we titrated single-labeled hAgo2$^{PAZ}$ with labeled guide RNA. Co-localization analysis of donor and acceptor dyes informs about the hAgo2 molecules that can be loaded by the addition of synthetic RNAs (presumably hAgo2 molecules that were not associated with cellular RNAs as RNA-guide complexes are generally extremely stable[46]). This resulted in an apo fraction of approximately 30% (Fig. S9B). Additional purification steps that aimed to liberate hAgo2 from cellular RNAs using heparin chromatography resulted in a hAgo2 preparation in which more than 60% of the hAgo2 molecules can be loaded with a synthetic guide RNA (Fig. S9C). Subsequent intra-molecular FRET measurements of the extensively purified hAgo2-$^{PAZ/Mid}$ apo enzyme also resulted in a broad distribution of FRET efficiencies with three distinguishable populations with mean FRET efficiencies comparable to the measurement using hAgo2 without a second purification step (compare Figs. 3A, S9A and Table S2). In case of heparin-purified hAgo2, however, more molecules are found in the population with a high FRET efficiency (Fig. S9D). In conclusion, unliganded hAgo2 shows a high degree of flexibility between the N-PAZ and the Mid-PIWI lobe. Upon guide RNA binding of hAgo2$^{PAZ/Mid}$ the three conformational states are still detectable (Fig. 3B). These states were found irrespective of whether hAgo2 was purified to be nucleic acid free or not. However, the fraction of molecules in the high FRET state decreased slightly compared to the apo enzyme. For intermolecular and intramolecular FRET measurements, we used a biotinylated target RNA for the immobilization of ternary complexes to ensure that only fully assembled ternary complexes are immobilized and included in these measurements. Upon association of the target RNA to binary hAgo2$^{PAZ/Mid}$/guide complexes, we could not detect any molecules that display a high FRET efficiency indicating that formation of the ternary complex results in a stabilization of hAgo2 (Fig. 3C). In case of the apo enzyme more than 20% of the molecules in the measurement dynamically switch between the conformational states (Fig. 3D). The fraction of molecules with dynamic behavior decreases upon guide and target RNA binding (Fig. 3D). A direct comparison of the FRET histograms of ternary complexes and the apo enzyme illustrates the shift of the molecule distribution from higher to lower FRET efficiencies upon target binding (Fig. 3E). Consequently, we suggest that

binding of nucleic acids stabilizes the hAgo2 lobes in an open conformation, which is in accordance with crystal structures[9]. However, our measurements also revealed the hitherto undetected flexible nature of hAgo2 even in nucleic acid bound states.

**Intra-lobal conformational heterogeneity enables the adaptation of human Ago2 to a variety of substrates.** To collect information about conformational changes along the intra-lobal axes, we examined conformations of the PAZ or the Mid domain in relation to residue F23, which marks a central pivot point between the two lobes of hAgo2. Because of its stable central position, we employed position 23 in combination with the labels in the PAZ or the Mid domain (hAgo2$^{N/PAZ}$ and hAgo2$^{N/Mid}$) to gain information about conformational changes of these domains within the N- or the C-terminal lobe, respectively (Figs. 4 and 5). FRET measurements using the apo enzymes revealed a single FRET population for each variant (Figs. 4A and 5A) with mean FRET efficiencies of 0.28 and 0.16, respectively. We observed two sparsely populated additional conformations for both hAgo2 variants upon addition of guide and target strands (Figs. 4A–C and 5A–C). Distance changes between the Mid domain and the N-terminal residue F23 are barely noticeable in crystal structures (Fig. S1B). Nevertheless, our measurements indicate a certain degree of conformational flexibility in between these positions. Structural information suggests an increase in distance between F23 in the N terminus and the labeled residue 291 in the PAZ domain. Interestingly, we found two populations with shorter distances upon guide and target loading. For the hAgo2$^{N/Mid}$ and the hAgo2$^{N/PAZ}$ variant, dynamic switching between all three populations (Fig. S10A, B) was observed. Dwell times and corresponding rate constants are comparable to data determined for other labeling positions (Figs. S7F,ii-G,ii, S9E and S10A, B) indicating that hAgo2 moves in a concerted fashion.

Finally, we tested miRNA451, which displays only seed complementarity to its target and little to no complementarity in the central, supplementary, and tail region of the guide RNA (Fig. 5D). Notably, the corresponding pre-miRNA is not processed by Dicer but by hAgo2[47]. Therefore, we additionally asked the question whether bulged pre-miRNA substrates influence the structural state of hAgo2 allowing efficient cleavage of this unusual pre-miRNA451 (Fig. 5E). Since the hairpin and the non-paired region of the guide-target duplex are located in

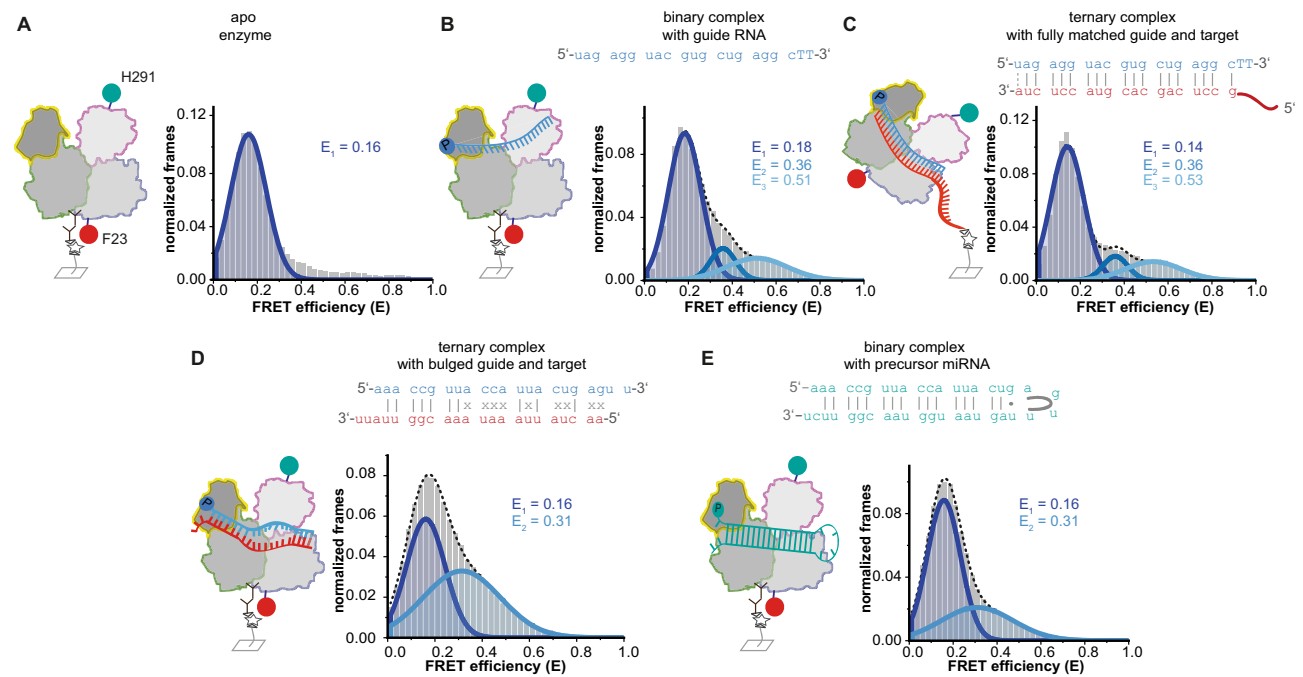

**Fig. 5 Conformational state of the N-terminal lobe upon binding of guide and target RNA.** Doubly labeled apo hAgo2^N/PAZ (**A**) and binary hAgo2^N/PAZ-guide RNA complexes (**B**) were immobilized via an hAgo2-directed antibody, whereas ternary hAgo2^N/PAZ-guide RNA-target RNA complexes (**C**) were immobilized via the target RNA^long-Biotin. In case of measurements with miRNA substrates used in (**D**) (guide RNA^p-miR451a and target RNA^osr1) and (**E**) (pre-miR451a) the hAgo2^N/PAZ complexes were immobilized using an hAgo2-directed antibody. All FRET efficiency histograms show data of at least three independent measurements. All data were fitted using a Gaussian equation. Source data provided as a Source Data file.

the 3'-region of the miRNA and therefore in the N-terminal lobe, we used the doubly labeled hAgo2^N/PAZ variant for smFRET measurements. Interestingly, upon binding of the pre-miRNA substrate with a hairpin loop at one side, we could detect only two populations (Fig. 5E) and the previously observed high FRET population (ternary complex: Fig. 5C, E = 0.53 and binary complex: Fig. 5B, E = 0.51) is not present. Similar data were collected for the guide/target combination with complementarity only in the seed region (Fig. 5D). To exclude sequence-specific effects, we also tested the miR451 guide alone, which led to results comparable to previous data collected with the siRNA guide (Figs. 5B and S10C). The high FRET population (E = 0.5) is disfavored in case of bulged or mismatched substrates (compare Fig. 5B–E). One explanation for this observation is that bulged substrates as well as the unpaired part of the target RNA demand space, which would prevent the PAZ domain from tilting over to engulf these substrates.

To test whether the 3'-bulge or the mismatched region causes the observed effects, we used a substrate that only contained a central bulge. In this case, we again found three populations congruent with data collected for the fully complementary target RNA (Fig. S10D) indicating that only supplementary mismatches and hairpins attached to the 3'-end of a miRNA disturb the movement in the N-terminal lobe.

## Discussion

Human Ago2 interacts with a multitude of protein and nucleic acid interaction partners[48,49]. Target RNA recognition and regulation requires a stable anchoring of guide and target RNA, but at some point, substrate exchange is necessary to proceed in the catalytic cycle. In this study, we followed the conformational status of hAgo2-RNA complexes, uncovered hitherto unseen conformational states of hAgo2 and described the transient and dynamic behavior of hAgo2-containing RNPs (Fig. 6).

**Conformational flexibility is a pre-requisite for Argonaute function.** In the apo state, the hAgo2 lobes switch between open and closed states, which possibly facilitates binding to nucleic acids as e.g. supplementary pairing of an RNA duplex is only possible upon opening of hAgo2[9]. Interestingly, also in case of apo Ago2 from *Drosophila melanogaster* (DmAgo) a broad range of FRET efficiencies has been detected, although in this case most molecules were found in a closed conformation[50]. The closed state of the two hAgo2 lobes is probably a state, which impairs the binding of nucleic acids. At this point additional proteins, for example Hsp chaperones, might assist hAgo2 to adopt a conformation that is more favorable for guide binding. It has been demonstrated that Hsp chaperones play an important role in stabilizing an open form of the DmAgo2 enzyme[50]. Since a significant fraction of hAgo2 molecules are found in the open conformation, we hypothesize that human Ago2 is not strictly dependent on chaperones for loading of nucleic acids. Stabilizing the open conformation of hAgo2 by Hsp chaperones might e.g. be a regulation mechanism to increase guide loading onto hAgo2 under certain conditions.

However, conformational flexibility is not only restricted to the apo state of hAgo2. Especially in ternary complexes, we found different positions of guide and target RNAs within hAgo2. A single structural state of the ternary complex could be detrimental to function by imposing an energetic barrier that hinders the progression in the activity cycle. In line with this idea, we found a large fraction of dynamic molecules in fully matched ternary complexes with the potential for catalytic activity (Fig. 6). This dynamic switching of the supplementary region relative to the PAZ domain in fully matched complexes, is explained by limited contacts of hAgo2 to the supplementary duplex and the finding that central base pairing is energetically disadvantageous in case of hAgo2[9,20]. Our data thereby extend the structural view on the conformational states of ternary complexes as our solution measurements show that ternary complexes exist not in a defined

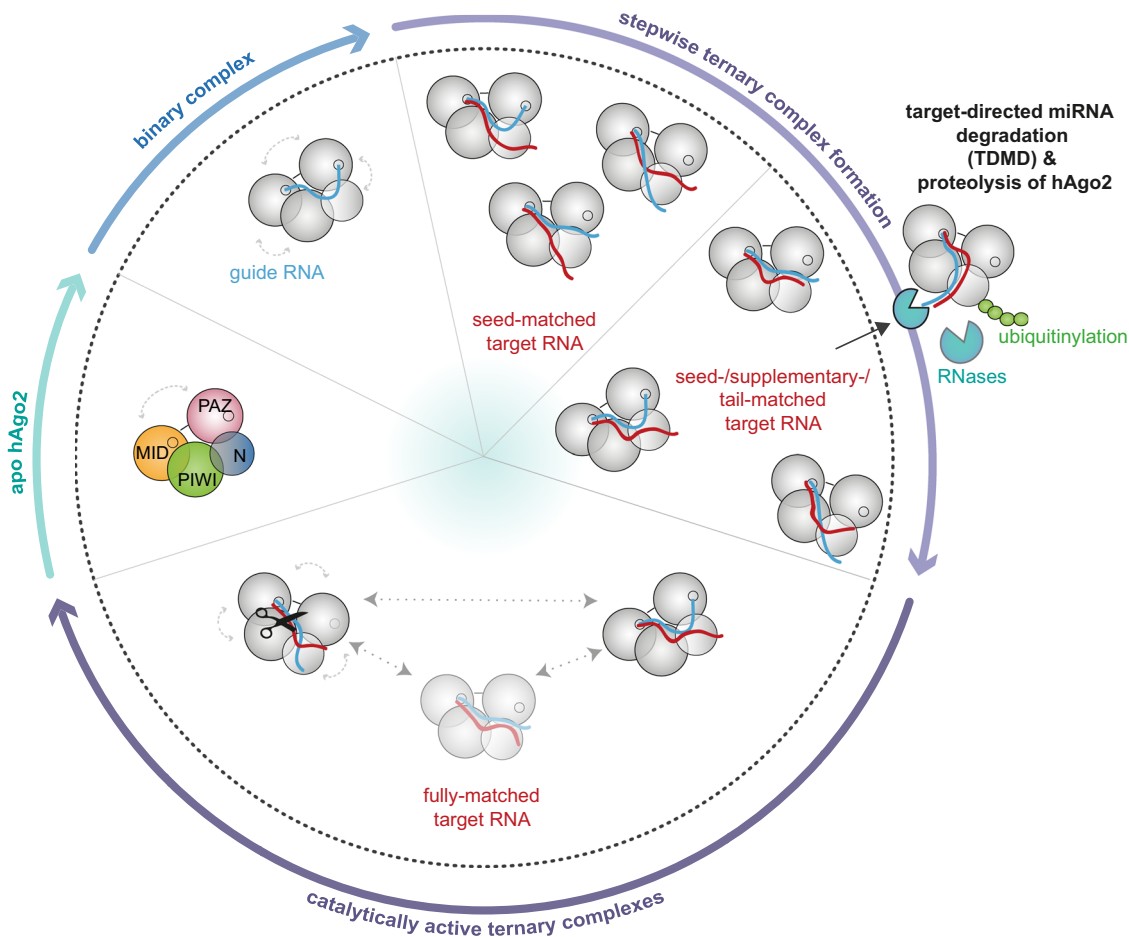

**Fig. 6 Schematic representation of the conformational states of human Ago2-RNA complexes based on single-molecule FRET studies in solution.** The hAgo2 apo state is characterized by extensive opening and closing motions of the Mid-PIWI and PAZ-N terminal lobes. Upon stable anchoring of the guide RNA within the Mid and the PAZ domain, the interlobal flexibility of the lobes decreases. Binary complexes are conformationally defined, but ternary complexes show a high degree of conformational heterogeneity. Stepwise association of the target is accompanied by conformational changes in hAgo2 and the guide and target RNA. Dependent on the extent of complementarity between guide and target RNA, different conformational states are adopted. Some of these states require the release of the 3'-end of the guide from the PAZ domain. Target RNAs with extensive complementarity but without central matches to the guide RNA adopt a conformation with an efficient exposure of the guide 3'-end. This conformational state is potentially coupled to an exit of this complex from the regulative cycle to enter the target-directed miRNA degradation pathway. Complexes composed of hAgo2 and a fully matched guide-target duplex are not found in a single but a variety of conformations including the catalytically active form (indicated by the scissor). The nucleic acids within these complexes frequently switch in between conformations. In addition, the domains along the axes in between the F23 and the Mid or the PAZ domain are mobile suggesting an engulfment of the bound RNA substrates. The shaded conformation is only transiently sampled. The 5'-end and 3'-end of the guide are anchored in the Mid and PAZ domain, respectively (anchoring points are indicated by small circles). Flexibility of hAgo2 domains or lobes are indicated by gray arrows.

single state. As suggested by structural and biochemical studies[19,26], our measurements also confirm that the individual strands of the supplementary RNA duplex are conformationally flexible. Nonetheless, a catalytically active conformation can be adopted and an exact positioning of the nucleic acids for target cleavage is assisted by intensive contacts to the 5'-half of the duplex[9] as well as functionalities of the N-terminal domain[17,18,51].

In contrast to the dynamic behavior of guide and target RNAs, our measurements disclose that the dynamics within the lobes decrease upon ternary complex formation. Hence, the inter-lobal opening and closing movements are not connected to the dynamic rearrangements of guide and target RNA in fully matched ternary complexes. Additionally, the conformational changes of the lobes themselves are uncoupled from inter-lobal conformations. In the apo enzyme, these intra-lobal axes are stiff as reflected by only a single conformation. Upon binding of guide and target, two additional states of the lobes appear, which induce

changes in the width of the nucleic acid binding channel. Ming et al.[52] assume that conformational changes like these can increase specificity, since cognate base pairs might be stabilized, whereas for non-cognate base pairing the opposite effect is induced. Analysis of substrates with bulges in the supplementary region reveal a restriction of motions in the N-lobe. In summary, we conclude that the conformational dynamics found in this study are a pre-requisite to allow binding and release of the nucleic acid substrates that are required for hAgo2 function.

**Implications for miRNA turnover.** In contrast to the dynamic behavior of guide and target RNAs within ternary complexes, the binary hAgo2-guide complex adopts a single conformation. This is consistent with the fact that guide strands bound to Ago are protected against degradation as anchoring of the 5'-end and the 3'-end in the Mid and PAZ domain, respectively, secludes them from nucleases[53]. However, upon formation of catalytically active

ternary complexes the guide 3'-end is released from the PAZ domain[10,26]. Jung et al. demonstrated that the interaction of the PAZ domain and the guide's 3'-end has an important function in target release[42]. While it has been assumed that 3'-end release depends on the extent of complementarity between guide and target, our data show (i) that 3'-end released and 3'-bound conformations co-exist during all stages of the target binding process and (ii) that the repetitive association and dissociation of the guide's 3'-end from the PAZ domain is possible, especially, if also the supplementary and central regions have the capability for base-pairing. Upon release from the PAZ domain, the guide's 3'-end is unprotected and free for enzymatic attacks[10,27,53], which could result in TDMD[10,31,32]. Nonetheless, hAgo2 is a multiple turnover enzyme that uses a single siRNA guide to cleave many complementary targets[54] and regulates targets with only small amounts of a certain siRNA or miRNA, since the presence of target RNA excess enhances the hAgo2 turnover number[55]. An instant decay of the guide upon release of the guide's 3'-end would therefore counteract hAgo2-mediated regulation, especially, if TDMD involves a proteolytic digest of the Ago protein[33,34]. An hAgo2-bound guide can be stable over days, whereas in some cases the half-lifes are very short, enabling rapid adaptation to changing conditions[27,56,57]. If the guide RNA is re-used in further regulation cycles, a preservation is necessary. In some organisms, methylation of the 2'-terminal oxygen protects miRNAs and siRNAs against degradation[31], which is, however, not the case for human miRNAs or siRNAs. In case of fully matched ternary complexes, we observe a repetitive re-attachment of the guide's 3'-end to the PAZ domain, which likely plays a role in protecting a hAgo2-bound guide against degradation by cellular nucleases. Interestingly, our measurements reveal that a seed-matching target also leads to these two conformations. There are many targets that are solely regulated by seed-pairing guides, which would result in the release of the guide's 3'-end. Hence, we hypothesize that the position of the guide's 3'-supplementary region has to differ in case of TDMD-inducing targets. Otherwise, all 3'-end released conformations could induce an efficient decay of the guide strand via the TDMD mechanism. Our smFRET data corroborate this hypothesis, because targets with central mismatches and otherwise excessive complementarity, so-called TDMD-inducing targets[30], exclusively lead to a population with an extraordinarily low FRET efficiency not detected for any other guide-target duplex (Fig. 6). Hence, we propose that centrally mismatched ternary complexes force the guide's supplementary region into a position, which likely pre-destines it for modification and degradation by nucleases implicated in TDMD[9,30]. A crystal structure of a ternary complex with a centrally mismatched target RNA supports this idea[10]. In this complex, no contacts between hAgo2 and the supplementary duplex are detectable. Moreover, the central mismatch induced a kink in the central region potentially forcing the supplementary duplex in between the N-terminal and the PIWI domain. Hence, the potential for central base pairing is decisive for the fate and stability of guide strands as well as the hAgo2 protein itself[33,34] upon target binding. The conformation determined in our measurements reflects a ternary complex state in which the guide (and possibly the guide-target duplex) is positioned further away from the PAZ domain when compared to the conformation found by Sheu-Gruttadauria et al.[10] Possibly, this conformation only occurs in solution and represents the conformational state that most efficiently triggers TDMD. Our data therefore rationalize the observed differences in guide RNA lifetimes that arise from different 3'-end release probabilities of TDMD targets and targets with full base pairing potential[10,30].

In summary, our smFRET study leads to a comprehensive model of hAgo2-mediated guide-dependent target binding that

revealed the different conformations that hAgo2 adopts in solution and that are required for hAgo2-based regulatory activity.

## Methods

**Generation of expression plasmid.** We used the FLAG-eGFP expression plasmid for hAgo2 generated before[37] based on Addgene plasmid #21981. Amber stop codons were introduced using the QuikChange lightning kit (#210518; Agilent).

**Cell culture.** Dulbecco's Modified Eagle Medium (DMEM) with high glucose (4.5 g/L) buffered with 25 mM HEPES without phenol red (#21063045; Thermo Fisher Scientific) supplemented with 1% (v/v) Penicillin/Streptomycin (10,000 U/mL) (#15140122; Thermo Fisher Scientific) and 10% (v/v) Fetal Bovine Serum (#F7524; Sigma Aldrich) was used to grow HEK293 T cells under standard conditions (37 °C/ 5 % $CO_2$). Cells were passaged every 2 – 3 days. After washing with PBS (137 mM NaCl, 2.7 mM KCl, 10 mM $Na_2HPO_4$, 2 mM $KH_2PO_4$), cells were detached using 1 mL/10 cm plate Trypsin (2.5%) (#15090046; Thermo Fisher Scientific). Appropriate amounts of cells were transferred to a new dish containing 10 mL fresh growth medium.

**Transfection of cells.** Transfection of cells was conducted as described before[37]. In brief, cells were seeded in 6-well or 10 cm cell culture dishes (#83.3920 or #83.3902; Sarstedt) to yield 60 - 70% confluency on the day of transfection. For transfection, jetPrime® (#114-15; Polyplus Transfection®, VWR) was used according to manufacturer's instructions with 4 µL jetPrime® reagent for 2 µg of DNA in a 6-well plate. Total amounts of transfected DNA were 2 µg, of which 1 µg were the hAgo2 expression vector. For the incorporation of p-Azido-L-phenylalanine (AzF) (#06162; ChemImpex International), 0.9 µg of the orthogonal Tyr-tRNACUA (tRNA fusion construct from *H. sapiens* and *B. stearothermophilus*) and 0.1 µg Tyr-aminoacyl-tRNA synthetase from *E. coli* encoded on different plasmids were co-transfected (plasmids supplied by T. Huber, Rockefeller University New York, USA[58,59]). At the day of transfection, an AzF stock (100 mM in 15% DMSO (#A994.1; Roth)) was diluted to 25 mM with HEPES/KOH, pH 7.5. Before transfection the medium was exchanged to new medium containing 200 µM AzF. Cells were harvested after 48 h and in case of 6-well plates, 2 wells were combined to yield a single cell pellet and stored at −80 °C until use. For evaluation of the expression efficiency, fluorescence microscopy of the living cells in the cell culture wells was conducted (EVOS® FL Cell Imaging System (Thermo Fisher)). For acquisition of fluorescence images, the GFP filter was used.

**Purification and labeling of modified protein.** For purification and labeling of hAgo2 protein[60] 750 µL of IP-lysis buffer (25 mM Tris-HCl pH 7.5, 150 mM NaCl, 1% NP-40 supplemented with cOmplete™ protease inhibitor (#589297001; Roche)) was used for resuspension of the cell pellets. Subsequently, resuspended cells were incubated on ice for 20 min and then centrifuged at 20.000 x g at 4 °C for 20 min to remove cell debris. The clarified supernatant containing extracted protein was incubated with 70 µL of ANTI-FLAG® M2 beads (#A2220-10 mL; Sigma Aldrich) (equilibrated in FLAG-1 buffer (50 mM Tris-HCl, pH 7.5, 200 mM NaCl, 0.05% NP-40)) for 1 h at 15 °C, shaking at 900 rpm in 1.5 mL reaction tubes. Three washing steps with FLAG-1 buffer were conducted to remove unspecifically attached proteins. Coupling of the fluorescent dyes (phosphine derivative of DyLight™550 or 650 (#88910 or #88911; ThermoFisher Scientific)) was performed in 100 µL FLAG-1 buffer using a final dye concentration of 100 µM. After addition of the dye to the hAgo2-bound beads, the beads were incubated for 2 h at 25 °C, shaking at 900 rpm, followed by three washing steps with FLAG-2 buffer (FLAG-1 + 2% (v/v) Tween-20 (#P-1379, Sigma Aldrich)) to remove excessive dye. Subsequently, the beads were transferred into a fresh tube using FLAG-1 buffer. To form binary complexes, the beads were incubated with 45 µL FLAG-1 buffer, 1.5 µL RiboLock (#EO0381;Thermo Fisher) and 3.5 µM guide RNA[14Cy5] or 7 µM guide RNA for 15 min, 25 °C and 900 rpm followed by three washing steps with FLAG-1 buffer. To form ternary complexes, the beads were incubated with 45 µL FLAG-1 buffer, 1.5 µL RiboLock and 7 µM target RNA or 3 µM target RNA[4Cy3] for 15 min, 25 °C and 900 rpm followed by three washing steps with FLAG-1 buffer. hAgo2 was eluted from the beads by incubating the beads with FLAG-1 buffer supplemented with 300 µg/µL 3xFLAG peptide (#F4799-25 mg; Sigma Aldrich) for 30 min at 25 °C, shaking at 900 rpm. The supernatant was used directly for single molecule experiments. Part of the sample was kept for SDS PAGE analysis followed by a fluorescence scan to ensure successful purification of labeled full-length FLAG-eGFP-hAgo2[AzF*DyLight].

To reduce the amounts of nucleic acids that co-purify with hAgo2, 1.5 mL of IP-lysis buffer was used to resuspend cell pellets from 10 cm cell culture plates. Subsequently, resuspended cells were incubated on ice for 20 min and then centrifuged at 20.000 x g at 4 °C for 20 min to remove cell debris. The clarified supernatant was incubated with 150 µL of ANTI-FLAG® M2 beads (equilibrated in FLAG-1 buffer) for 1 h at 15 °C, shaking at 900 rpm in 1.5 mL reaction tubes. Three washing steps with FLAG-1 buffer were conducted to remove non-specifically attached proteins. Coupling of the fluorescent dyes (phosphine derivative of DyLight™550 or 550 and 650) was performed in 200 µL FLAG-1 buffer using a

final dye concentration of 100 μM. After addition of the dye to the hAgo2-bound beads, the beads were incubated for 2 h at 25 °C, shaking at 900 rpm, followed by three washing steps with FLAG-2 buffer (FLAG-1 + 2% (v/v) Tween-20) to remove excessive dye. Subsequently, a washing step with heparin washing buffer (50 mM Tris pH 7.5, 50 mM NaCl) was performed. Labeled hAgo2 was eluted from the beads by incubating the beads with heparin washing buffer supplemented with 300 μg/μL 3xFLAG peptide for 30 min at 25 °C, shaking at 900 rpm. The supernatant was directly applied to a heparin column (HiTrap® Heparin High Performance, 1 mL, #GE17-0406-01; Cytiva, equilibrated in heparin washing buffer) to separate apo hAgo2 and nucleic acid bound states using the Azura FPLC system (Knauer). After a washing step (5 column volumes (cv) with heparin washing buffer, elution of bound proteins was performed using an elution gradient (8 cv, buffer a = heparin washing buffer, buffer b = heparin elution buffer (50 mM Tris pH 7.5, 1 M NaCl)). Selected hAgo2-containing elution fractions were incubated with 60 μL of ANTI-FLAG® M2 beads (equilibrated in FLAG-1 buffer) for 30 min at 25 °C, shaking at 900 rpm in 1.5 mL reaction tubes. This was followed by three washing steps with FLAG-1 buffer. Afterwards, the samples were split and incubated with or without guide RNA in 30 μL FLAG-1 buffer including 1.5 μL RiboLock. After three washing steps with FLAG-1 buffer, the samples were eluted in FLAG-1 buffer supplemented with 300 μg/μL 3xFLAG peptide for 30 min at 25 °C, shaking at 900 rpm. The supernatant was directly used for single molecule experiments. hAgo2 preparations that were purified including this additional heparin step were used for smFRET measurements shown in Fig. 3A, B and S9C–E.

**Single molecule pulldown experiment**. Purified and labeled samples were applied to custom-built flow chambers[61]. Before applying the sample, the flow chamber was cleaned and incubated for 5 min with 0.1 mg/mL NeutrAvidin (#31000; ThermoFisher Scientific) in FLAG-1 buffer. To remove excessive NeutrAvidin the chamber was flushed with FLAG-1 buffer afterwards. Subsequently, the biotinylated secondary antibody (rabbit-anti-rat, #31834; Invitrogen (for primary antibody 11A9) or rat anti-mouse, #13-4013-85, eBioscience (for MsmAb to Argonaute-2, #ab57113, abcam)) was applied to the flow chamber to decorate the surface with antibody using a biotin-NeutrAvidin linkage. The secondary antibodies were diluted 1:200 (rabbit anti-rat, Invitrogen) or 1:100 (rat anti-mouse, eBioscience) in FLAG-1 buffer. After 15 min incubation the flow chamber was again flushed with FLAG-1 buffer and afterwards incubated with the hAgo2-directed antibody (11A9[39] (1:200 in FLAG-1 buffer) or MsmAb to Argonaute-2, abcam, (1:100 in FLAG-1 buffer)). Again, after 15 min incubation time, excessive antibody was removed by flushing with FLAG-1 buffer.

After this procedure the sample was applied to the surface. The dilution of sample was conducted in FLAG-1 buffer and was depending on the expression efficiency of the proteins used. The incubation varied between 1 and 5 min. After incubation, the flow chamber was flushed with FLAG-1 buffer supplemented with an oxygen scavenging system (1% (w/v) glucose, 100 U/mL glucose oxidase (#G2133-10KU, Merck), 160 U/mL catalase (#C9322, Merck) and 5 mM of Trolox (dye photo stabilizer; #238813, Sigma-Aldrich)).

**Widefield single molecule detection and analysis**. Single-molecule FRET experiments on immobilized samples were carried out with a homebuilt prism-type total internal reflection (TIRF) microscope setup based on a Leica DMi8 inverse research microscope as describe earlier[37].

Samples were excited at 532 nm or 561 nm using solid state lasers (OBIS LS; Coherent) or diode laser at 637 nm (OBIS LX; Coherent, clean-up filter ZET 635/10; Chroma Technology). Excitation laser beams were combined using dielectric (BB1-E02; Thorlabs) and dichroic mirrors (ZT 532RDC, ZT 568LPXR superflat; Chroma Technology). In all cases, alternating laser excitation (Multistream; Cairn Research) was employed[62].

Fluorescence light was collected using a Leica HC PL Apo 63x N.A. 1.20 water-immersion objective and spectrally separated into donor and acceptor detection channels (Optosplit III; Cairn Research, see supplementary table 3 for sample-specific filters). Both detection channels were recorded simultaneously by an EMCCD camera (iXon Ultra 897, EM-gain 20, framerate 10 or 20 Hz; Andor Solis Version 4.31; Andor Technology).

The iSMS software package[63] was used for data analysis. Donor and acceptor fluorescence spots were selected based on a signal-to-background threshold using the automated spot finder. Individual traces were background-corrected by subtracting the average intensity of all pixels in a two-pixel distance to the fluorescence spot. Correction factors accounting for differences in fluorophore quantum yield and detection efficiencies (γ factor), donor leakage ($c_{leak}$) and direct excitation of the acceptor($c_{dir}$) were calculated from individual fluorescence-time traces using the functions implemented in iSMS. The mean of these individual correction factors was applied for the global correction of the FRET efficiency $E$ according to the equation:

$$E = \frac{I_{DA} - (c_{leak} \cdot I_{DD} + c_{dir} \cdot I_{AA})}{\gamma \cdot I_{DD} + I_{DA} - (c_{leak} \cdot I_{DD} + c_{dir} \cdot I_{AA})} \qquad (1)$$

with $I$: background-corrected fluorescence intensity. Subscript DD denotes donor emission after donor excitation, DA denotes acceptor emission upon donor excitation, AA denotes acceptor emission upon acceptor excitation.

Further data analysis was performed with Origin 2019 (Origin Lab). FRET efficiency histograms for individual experiments were calculated from all frames before the first bleaching event using a bin size of 0.025 (40 bins). Histograms were normalized to the total number of frames and the mean of each bin was calculated for all biological replicates to generate the final histogram. The number of replicates is given in the respective figure legends and in Supplementary Table 2. This table also includes standard deviations, fitting errors, and molecule numbers for all measurements. The mean FRET efficiency of detected conformational states of hAgo2/hAgo2-RNA complexes was determined by fitted the FRET efficiency histograms with a single or multimodal Gaussian equation. We generally used the minimum number of Gaussians that fit the overall distribution well ($R^2$ at least 0.97). For individual histograms (see Supplementary Table 2) we applied a priori knowledge about the state distribution (i.e. measurements with a single Gaussian fit or data from dynamic FRET traces) to incorporate additional populations into the fit that were not resolved or underrepresented in the data.

FRET efficiency-time traces displaying dynamic switching between different FRET efficiencies were further analyzed to calculate the average dwell time in the respective FRET state. A Hidden Markov Model (vbFRET algorithm[64], integrated in iSMS) was used to fit periods of similar FRET efficiency within a trace to discrete states, corresponding to distinct conformations of the complex. Dwell time histograms for each state were fitted with a mono-exponential decay function to determine the mean dwell time τ in the respective state:

$$N(t) = N_0 + A \cdot e^{-\frac{t}{\tau}} \qquad (2)$$

with $N(t)$: the occurrence of dwell time $t$, A: the amplitude of the decay and $N_0$: the y-axis offset. Information on transitions were further visualized in a transition density plot displaying possible transitions observed in the experiment.

**Immunoprecipitation of hAgo2 for in vitro cleavage assays with radioactively labeled target RNA**. For in vitro cleavage assays a cell pellet from cells cultivated in a 10 cm petri dish was used. For immunoprecipitation of hAgo2 cell pellets were dissolved in 1 ml NET buffer (50 mM Tris/HCl, pH 7.5, 150 mM NaCl, 5 mM EDTA, 0.5% (v/v) NP-40, 1 mM NaF, 0.5 mM DTT, and 1 mM AEBSF) and incubated for 20 min. After 20 min centrifugation at 15.000 g at 4 °C cell debris was pelleted and the clear lysates were added to 40 μl packed volume of anti-FLAG® M2 agarose beads. Beads were washed twice with cold PBS and incubated for 2.5 h at 4 °C with the lysates. Beads were washed three times with wash buffer (50 mM Tris/HCl, pH 7.5, 300 mM NaCl, 5 mM EDTA, 0.5% (v/v) NP-40, 10% (v/v) glycerol, 1 mM NaF, 0.5 mM DTT, and 1 mM AEBSF) and once with PBS. Samples were split for in vitro cleavage assay (65%) and Western blot analysis (35%).

**hAgo2 activity assay using radioactively labeled target RNA**. For preparation of cap-$^{32}$P-labeled target RNA (Supplementary Table 1)[4], the 20 μl cap-labeling reaction (40 mM Tris pH 8.0, 6 mM DTT, 15 mM MgCl₂ (#A1101, Diagonal GmbH & Co. KG), 2 mM spermidine-trihydrochlorid (#A0674, AppliChem), 25 μM SAM (#A7007-5 mg, Sigma Aldrich), 10 U RiboLock RNase inhibitor, 2 μg guanylyl transferase, 2 μg target and 20 μCi α-$^{32}$P-GTP (Hartmann Analytic)) was incubated for 4 h at 37 °C. After adding 2x formamide loading dye, the radioactive target was loaded onto an 8 % urea gel (sequencing gel diluent Rotiphorese®, #3047.1, Roth). Electrophoresis was performed for 1 h at 400 V. The RNA was visualized by UV shadowing, cut and eluted (300 mM NaAc, 1 mM EDTA, 0.1 % SDS) over night at 10 °C shaking at 300 rpm. The RNA was precipitated with 1 ml of 100 % ethanol for 30 min at 17 000 x g, 4 °C. After washing the pellet with 70 % ethanol, the pellet was airdried and dissolved in ddH₂O.

For in vitro cleavage reactions, translation mix was added to the beads to a final concentration of 1x (66.7 mM KCl, 6.7 mM MgCl₂, 8.3 mM DTT, 1.7 mM ATP, 0.3 mM GTP and 3.2 U RiboLock RNase inhibitor). The cleavage reaction was started by addition of 1-2 Bq/cm² cap-labeled target RNA and incubated for 60 min at 30 °C. The reaction was stopped by adding TRIzol® (#15596026; Thermo Fisher Scientific), and the RNA was extracted according to the manufacturer's protocol. After precipitation, the RNA pellet was dissolved in 2x formamide loading dye and it was separated on an 8 % sequencing gel (Rotiphorese®, Roth). For detecting the signals, the gel was dried, exposed to a phosphoimaging screen and scanned using a phosphoimager (PMI, Bio-Rad).

**Western blot analysis**. For Western blot analysis that reports on the presence hAgo2 in target cleavage reactions, samples were mixed with Laemmli buffer containing 10% β-mercaptoethanol and incubated at 95 °C for 5 min. After separation by SDS-PAGE, proteins were transferred to an AmershamTM Pro-tranTM 0.45 μm NC (#GE10600002; GE Healthcare) nitrocellulose membrane by semidry blotting. For protein identification we used the hybridoma supernatant containing monoclonal antibodies for human Ago2 (11A9[39]) (dilution 1:5 in 5% milk TBS-T buffer). For detection, a fluorescently labeled goat polyclonal anti-rat secondary antibody (IgG IRDye 800CW, 1:10000; #926-32219, Li-Cor Biosciences) was used and fluorescent signals were detected using the Odyssey Infrared Imaging System (LI-COR Biosciences).

For Western blot detection of hAgo2 abortion products, samples were mixed with Laemmli buffer with a final concentration of 2% (v/v) β-mercaptoethanol and

2% (w/v) SDS and incubated at 95 °C for 5 min. After separation by 10% SDS-PAGE, proteins were transferred to a Nitrocellulose 0.45 µm membrane (#GE10600002; Cytiva) by semidry blotting. For protein identification, we used purified monoclonal antibodies directed against human Ago2 (11A9[39]; 1:500). For detection, a fluorescently labeled goat anti-rat secondary antibody (Alexa647, 1:5000; #A21247, Life Technologies) was used and fluorescent signals were detected using the ChemiDoc imaging system (Biorad).

**Reporting summary**. Further information on research design is available in the Nature Research Reporting Summary linked to this article.

## Data availability

The data generated in this study are available as a source data file, which is provided along with this paper. PDB files of hAgo2 crystal structures that were used in this study are found using the following PDB identifiers: 4W5N, 4W5T, 6MDZ, 6NIT, 6N4O. Remaining raw datasets are available from the corresponding authors upon reasonable request. Source data are provided with this paper.

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

## Acknowledgements

We gratefully thank Tobias Restle for fruitful discussions in the early phase of the project, who was a great mentor, enthusiastically studied the biochemical mechanism of hAgo2 and sadly passed away too early. We gratefully acknowledge financial support by the Deutsche Forschungsgemeinschaft (GR 3840/2-2 and SFB960-TP7 to D.G.). We would like to thank Thomas Sakmar for providing plasmids that allow the incorporation of unnatural amino acids in human cells. Furthermore, we would like to thank Elisabeth Piechatschek and Elke Papst for technical assistance. We thank Alexander Gust for experimental support in the early stages of this project.

## Author contributions

D.G. and S.W. conceived the study. S.W. and L.J. performed cell culture experiments. S.W. and L.J. performed the single-molecule measurements. S.W. and L.J. analyzed the single-molecule data. V.G. and J.N. performed cleavage assays. G.M. designed and oversaw cleavage assay experiments. S.W. and D.G. wrote the paper. K.K. assisted with single-molecule FRET analysis and wrote part of the methods section. All authors commented on the paper.

## Funding

## Competing interests

The authors declare no competing interests.
