## [Peer Review File · Nature Communications]

Title: Single-molecule FRET uncovers hidden conformations and dynamics of human Argonaute 2REVIEWER COMMENTS

Reviewer #1 (Remarks to the Author):

Willkomm and co-workers present a fascinating and important study of human Ago2 dynamics using single molecule FRET. To my knowledge, this is the first work of its kind on human Ago2 and the study provides expected findings relevant to understanding small RNA biology and, potentially, to advancing development of therapeutic siRNAs. The work is a welcome addition to the field, which has many static Ago2 crystal structures available, but far less insight into conformational dynamics. Major findings include: dynamic anchoring and release of the guide 3'-end, even upon seed-pairing and extended target binding, a previously unknown conformation that may be associated with target-directed miRNA degradation, and, potentially, insights into Ago2 conformational changes associated with guide loading. I am very excited about the work, but have a few major concerns and many small comments/suggestions:

Major concerns:

1) A major concern surrounds the quality of apo Ago2 preparations used for experiments in Fig. 4 and 5. The methods do not indicate any special procedure for specifically preparing apo Ago2 or demonstrating that the samples are free of co-purifying RNAs. Considering ectopically expressed Ago proteins in cell culture associate with miRs (PMID: 15260970), and a fraction of molecules in the Ago2 preparations of this study were shown to co-purify with endogenous miR-19b (Fig. S3B), it is reasonable to expect that the samples used to examine apo Ago2 also contained endogenous miRs. It is not clear how the authors can differentiate apo Ago2 FRET signal from signal arising from Ago2 molecules that co-purified with endogenous miRs, especially without knowing the fraction of molecules that are apo in their sample. The authors should verify that their apo samples are indeed not contaminated with co-purifying miRs or other RNAs. This could be done by measuring absorbance as a protein sample free of contaminating nucleic acid should have an A260/A280 ratio equal to 0.5. A ratio >0.5 would indicate co-purifying RNA. If present, the relative amount of co-purifying RNA could be assessed by absorbance using calculated extinction coefficients, or by denaturing gel (with RNA standards of known concentration). Additionally, the authors could report the fraction of "loadable" Ago2 in their samples. miRs dissociate so slowly from Ago2 that the binding is nearly irreversible (PMID: 23664376). Thus, the loadable fraction may be considered the upper limit for the fraction of Ago2 molecules that are apo in the sample. This can be done by observing the fraction of Ago2 pulled down by an immobilized target oligo after loading with their guide RNA (see PMID: 23249751). Alternatively, this information might more easily be obtained by examining fluorescence of samples like those used in Fig. 1C—by comparing the number of total DyLight550 spots (indicating an Ago2 molecules) to DyLight550 spots that also contain Cy5 fluorescence (indicating bound guide RNAs).

2) A related concern surrounds the apparent lack of purification of Ago2 molecules loaded with specific guide and target RNAs for experiments in Fig. 4 and 5. If a substantial fraction of Ago2 molecules in the preparation are bound to endogenous miRs, and potentially co-purifying target RNAs, the changes in FRET data presumed to be associated with target binding would be much more difficult to interpret. Again, assessing the quality of Ago2 samples, with respect to removal of co-purifying endogenous RNAs,

would assuage these concerns and substantially strengthen the conclusions.

3) It is curious that the apo and binary complex FRET signals in Fig. 3 are so similar, considering analogous experiments using *Drosophila* Ago2 produced obviously different results (PMID: 29775584). It would be surprising and interesting to learn that human and fly Ago2 have such different conformational changes associated with guide binding. However, considering my concerns above, I not yet fully convinced this is true.

4) The section describing results in Fig. 5 is confusing and seems illogical at points. For example: “Since bulges and the hairpin are located in the 3’-region of the miRNA and therefore in the N-terminal lobe.” This reasoning seems incongruent with the structure of pre-miR451 in Fig. 5E, which has no notable bulges (only a G:U wobble pair at the very end of the duplex). And, “Similar data were collected for the guide/target combination with two bulges in the central and supplementary region of the guide (Figure 5D).” The guide and target in Fig. 5D have so little complementarity after the seed that I do not believe it is possible to reliably predict the structure of any potential duplex that might form (especially while bound to Ago2).

Minor comments/concerns

5) Line 163/Fig 2C: The 0.66 nm distance larger than that observed in crystal structures in ref 10 is intriguing. Might it be connected to the observation that the RNAs in these crystal structures were held in place by crystal lattice contacts? Indeed, in one structure (PDB ID: 6MFN, see Fig. 1b in ref 10), which lacked lattice contacts, the portion of RNA containing g14 was completely disordered. I am therefore wondering: does the E1 peak in Fig 2C correspond to a well-defined conformation that could be observed via structural methods (but has gone unobserved so far), or does the E1 peak arise from a family conformations that resemble structures in ref 10 (in terms of protein conformation), but are dynamic (in terms of RNA conformation) and thus be represented by PDB ID: 6MFN?. Can the data distinguish between these two possibilities?

6) Statement on lines 97-98 should include citation to PMID: 26140367. It would also be reasonable to cite PMID: 26140592, as mouse and human Ago2 are nearly identical.

7) Line 128 “seed-matching target (g1 – g8)” should be “seed-matching target (g2 – g8)” as g1 is not part of the seed and does not physically match with the target. This minor error occurs multiple other places in the manuscript.

8) Lines 137-144. The wording here carries a sense of absoluteness that is inaccurate. For example, “...according to structural studies, the 3’-end of the guide is still anchored in the PAZ domain...” indicates crystal structures provide a full view of the RNA-protein complex, which of course is not the case as crystal structures only provide snap-shots of conformations that are low energy enough to enable crystallization. Similarly, “...base pairing within the seed region is sufficient to trigger the release of the 3’-end from the PAZ domain...” does not reflect the observation that E3 remains the most populated state in the presence of the seed-matched target RNA. Perhaps “...is sufficient to stimulate the

release...." is more accurate?

9) Fig. 2: why does the distribution of E3 broaden in panels C and D (compared to A and B)? (this is not a criticism, just curiosity).

10) Line 195-196: "Although crystal structures indicate that a centrally paired catalytically active ternary complex is relatively unstable (ref 9)". This statement is inaccurate. Ref 9 only noted that even when central pairing is available, Ago2 molecules can maintain a stable conformation in which central pairing is avoided. This is not evidence that a centrally paired catalytically active ternary complex is unstable or even less stable.

11) Line 368: "Our data therefore rationalize the observed differences in guide RNA lifetimes that arise from different 3'-end release probabilities of TDMD targets and targets with full base pairing potential." This statement needs a reference.

12) Line 29-30 "which ultimately leads to hAgo2-catalyzed degradation of the mRNA or translational inhibition" Consider "hAgo2-mediated degradation" or "hAgo2-directed degradation", as catalytic cleavage only works with special targets/precursors or artificial guides, while likely the majority of transcripts targeted by Ago2 are catalytically degraded by other enzymes.

13) Line 77-78: "Binding of TDMD targets is accompanied by the release of the guide's 3'-end exposing this end for modification...thereby possibly triggering the proteolysis of Ago." This could be worded more precisely, as ref 33 and 34 both propose trimming-tailing-independent degradation of Ago.

14) Figure 2 Legend E is missing.

Reviewer #2 (Remarks to the Author):

Willkomm and colleagues examined the conformational dynamics of hAgo2 using site-specifically labelled hAgo2 and single-molecule FRET. The results obtained by the authors confirm previous studies and reveal new conformational states. They observed a catalytically active conformation and also reported a conformational state that is unique for TDMD-inducing targets. Overall, this well-designed and well-executed study provides new and interesting insights into the conformational dynamics of hAgo2. Additionally, the manuscript is well written. Therefore, this reviewer enthusiastically recommends this manuscript for publication in Nature Communications.

SEP

Minor comments

1. They authors should describe how they determine how many Gaussians to fit to each histogram.

2. While FRET is sensitive to nanometer distance changes, it cannot be and should not be used to estimate an absolute distance. For example, the description in line 160 ("The large corresponding

distance of 6.6 nm suggests ...) should be moderated.

3. Description of Figure 2E is missing in the figure caption.

Reviewer #3 (Remarks to the Author):

The authors utilize a site-specific protein labeling approach (that they have published before) to introduce FRET labels into human Argonaute-2 (hAgo2). They then collectively use the labelled hAgo2 variants (complemented with labeled guide RNAs) to explore the structural versatility of apo-hAgo2 and to study its dynamic transitions upon guide RNA binding and target recognition in various experimental scenarios. Whereas parts of this study, especially regarding seed-target pairing, constitute primarily supportive evidence for prior results gained by structural biology, the study also reveals new, exciting insights into target-directed microRNA degradation, pre-miR451 processing and the cleavage-competent hAgo2 conformation. Provided that my concerns are adequately addressed, I see this study as an important addition to previous structural work on hAgo2 and therefore suitable for publication in Nature Communications.

Major points

1. The authors define the FRET label at hAgo2 residue F23 as being part of the PAZ-N lobe of hAgo2. This statement needs further rationalization (and explanation), as residue 23 of hAgo2 directly packs against a helix (675-694) in the PIWI domain of hAgo2. I therefore posit that the label is dynamically part of the MID-PIWI lobe, not the PAZ-N lobe. Given this discrepancy in interpretation, I urge the authors to rationalize their labeling choice. In particular, the authors need to experimentally clarify why the F23 label dynamically represents the N-lobe of hAgo2. Should this not be the case, the authors need to re-interpret their data throughout the entire study.
2. Figs. S3A and S3B do not convince me of the authors' statements that all hAgo2 eGFP fusion proteins are catalytically equally active and that the experimental conditions prevent target cleavage from occurring. More convincing biochemical evidence in support of the authors' statement is required here.
3. The observed significant kink in a TDMD-prone guide/target duplex is intriguing and an interesting result of this study. However, to generalize this finding with regards to the mechanism of TDMD, the authors should at least to use one additional miRNA/target RNA pair for their experiments.
4. For a fully matched guide/target complex (Fig. 2D) the authors find two distinct populations. The one with lower FRET efficiency (0.35) they interpret as the catalytically active conformation, without providing any experimental proof for this statement. As the authors refer to this low-FRET population throughout the manuscript as the "active" conformation, I suggest that the authors set out to provide experimental evidence that this conformation correlates with hAgo2 activity.

Minor points

1. As I found the results of their study more exciting than the presented abstract, I encourage the authors to consider rewriting parts of the abstract to better reflect their results.
2. The cartoons symbolizing the respective experimental FRET setup are crucial to aid understanding. If possible, their relative size (with respect to the entire figure) should be increased to make subtle differences visible.
3. Fig. S2B: The authors refer to low-molecular weight protein products as “abortion products”. What is meant by this? The doubly-labelled hAgo2 variants show significant impurities. Did the authors analyze the nature of these bands? Are protein yields sufficient to run Coomassie-stained SDS PAGE?
4. Fig.5: The authors refer to the miR-451-target interaction as possessing two bulges. As judged from the corresponding figure (Fig. 5D), I am unable to comprehend this statement.
5. Figures are not consecutively arranged according to the flow of the manuscript, e.g. Figs. 5D and 5E are being referred to after Fig. 5B is. I urged the authors to correct this to aid the reader.
6. How can the role of chaperones in hAgo2 loading and the authors’ FRET data on apo-hAgo2 be integrated? I missed any comments on this potential connection, e.g. in the discussion. The authors may want to include thoughts in this direction in their manuscript.

Point-by-point response:

We thank the reviewers for the detailed and careful evaluation of our study, the positive reception and constructive feedback that helped us to further improve our manuscript. We addressed all concerns raised by the reviewers, conducted additional experiments, and changed the manuscript accordingly. Please, find our point-by-point response below.

Reviewer 1

remarks to the author:

Willkomm and co-workers present a fascinating and important study of human Ago2 dynamics using single molecule FRET. To my knowledge, this is the first work of its kind on human Ago2 and the study provides expected findings relevant to understanding small RNA biology and, potentially, to advancing development of therapeutic siRNAs. The work is a welcome addition to the field, which has many static Ago2 crystal structures available, but far less insight into conformational dynamics. Major findings include: dynamic anchoring and release of the guide 3'-end, even upon seed-pairing and extended target binding, a previously unknown conformation that may be associated with target-directed miRNA degradation, and, potentially, insights into Ago2 conformational changes associated with guide loading. I am very excited about the work but have a few major concerns and many small comments/suggestions.

Major comments:

Comment:

A major concern surrounds the quality of apo Ago2 preparations used for experiments in Fig. 4 and 5. The methods do not indicate any special procedure for specifically preparing apo Ago2 or demonstrating that the samples are free of co-purifying RNAs. Considering ectopically expressed Ago proteins in cell culture associate with miRs (PMID: 15260970), and a fraction of molecules in the Ago2 preparations of this study were shown to co-purify with endogenous miR-19b (Fig. S3B), it is reasonable to expect that the samples used to examine apo Ago2 also contained endogenous miRs. It is not clear how the authors can differentiate apo Ago2 FRET signal from signal arising from Ago2 molecules that co-purified with endogenous miRs, especially without knowing the fraction of molecules that are apo in their sample. The authors should verify that their apo samples are indeed not contaminated with co-purifying miRs or other RNAs. This could be done by measuring absorbance as a protein sample free of contaminating nucleic acid should have an A260/A280 ratio equal to 0.5. A ratio >0.5 would indicate co-purifying RNA. If present, the relative amount of co-purifying RNA could be assessed by absorbance using calculated extinction coefficients, or by denaturing gel (with RNA standards of known concentration). Additionally, the authors could report the fraction of "loadable" Ago2 in their samples. miRs dissociate so slowly from Ago2 that the binding is nearly irreversible (PMID: 23664376). Thus, the loadable fraction may be considered the upper limit for the fraction of Ago2 molecules that are apo in the sample. This can be done by observing the fraction of Ago2 pulled down by an immobilized target oligo after loading with their guide RNA (see PMID: 23249751). Alternatively, this information might more easily be obtained by examining fluorescence of samples like those used in Fig. 1C—by comparing the number of total DyLight550 spots (indicating an Ago2 molecules) to DyLight550 spots that also contain Cy5 fluorescence (indicating bound guide RNAs).

A related concern surrounds the apparent lack of purification of Ago2 molecules loaded with specific guide and target RNAs for experiments in Fig. 4 and 5. If a substantial fraction of Ago2 molecules in the preparation are bound to endogenous miRNAs, and potentially co-purifying target RNAs, the changes in FRET data presumed to be associated with target binding would be much more difficult to interpret. Again, assessing the quality of Ago2 samples, with respect to removal of co-purifying endogenous RNAs, would assuage these concerns and substantially strengthen the conclusions.

It is curious that the apo and binary complex FRET signals in Fig. 3 are so similar, considering analogous experiments using *Drosophila* Ago2 produced obviously different results (PMID: 29775584). It would be surprising and interesting to learn that human and fly Ago2 have such different conformational changes associated with guide binding. However, considering my concerns above, I not yet fully convinced this is true.

Response:

Our experimental design is based on the fresh preparation and labeling of native hAgo2 variants from HEK cells, which is performed prior to every single molecule FRET experiment. This is on the one hand a strength of our study. On the other hand, it poses the possibility that the hAgo2 preparation used is pre-loaded with cellular RNAs. We agree with the reviewer that the analysis of for example the apo form of hAgo2 via smFRET measurements requires knowledge about the nucleic acid binding state of the molecules analyzed. Thus, we followed the suggestion of the reviewer and compared the number of molecules that show co-localization of donor (DL550-labelled hAgo2^{PAZ}) and acceptor (Cy5-labelled guide RNA) dyes to the total number of donor molecules. This analysis revealed that 30 % of all hAgo2 molecules can be loaded with an RNA guide (new Figure S9A-B). We aimed to increase the number of nucleic acid free hAgo2 molecules in our preparations, which is a challenging task as hAgo2 gets unstable in its apo form. To this end, we tested different approaches including cation exchange chromatography and heparin affinity chromatography. While these approaches required substantial amounts of cells, the heparin affinity chromatography purification was successful. Co-localization studies revealed that approximately 60 % of the molecules can be loaded with a synthetic RNA guide. Therefore, we performed smFRET measurements using heparin-purified hAgo2^{PAZ/Mid} in its apo as well as in its guide-bound state (new Figure 3A and 3B and S9). We still detect a broad range of FRET efficiencies especially in the apo enzyme but notice a change in the FRET efficiency distribution compared to the smFRET measurement of apo hAgo2^{PAZ/Mid} purified using our standard protocol. The medium FRET population decreased in favor of the high FRET population (new Figure S9D). While we did not succeed in the preparation of hAgo2 exclusively in its nucleic acid free apo form, the new measurements emphasize the tendency of apo hAgo2 to adopt a more closed conformation, which we already stated in our earlier interpretation. This agrees with data collected with *Drosophila* Ago2 (Tsuboyama et al., 2018). FRET efficiency histograms of binary complexes composed of hAgo2^{PAZ/Mid} and guide RNA (Figure 3B) also display these three populations, although the distribution of the molecules shifts to lower FRET efficiencies compared to the apo enzyme. These tendencies were also detected in *Drosophila* Ago2, although with a significantly stronger populated high FRET population in case of the apo enzyme. In comparison to *Drosophila* Ago2, hAgo2 displays a higher flexibility in the apo form as well as in the guide-bound state.

Intramolecular FRET measurements of hAgo2^{N/PAZ} and hAgo2^{N/Mid} in the apo state (Figure 4A and Figure 5A) resulted in defined single FRET populations. In contrast, measurements of binary complexes of these variants showed additional FRET populations. Nonetheless, the respective “apo” state population is still prominent indicating that this conformation is frequently sampled by hAgo2-guide and hAgo2-guide-target complexes.

Regarding the concern of the reviewer that in case of ternary hAgo2-guide-target complexes only a fraction of the complexes is actually associated with a target RNA: our original study design accounts for that as we used target RNAs with a biotin label at the 5'-end (as compared to binary complexes and apo hAgo2, we did not use hAgo2-directed antibodies for immobilization of ternary complexes). Using the biotinylated RNA, exclusively ternary complexes will be immobilized on the TIRF surface and used for smFRET measurements.

Changes:

We included new data collected using the heparin affinity chromatography purified hAgo2 variants including measurement of hAgo2^{PAZ}-guide RNA^{14Cys}, hAgo2^{PAZ/Mid}-guide RNA complexes and hAgo2^{PAZ/Mid} apo enzyme in the revised version of the manuscript (Figure 3) and the Supplementary data (Figure S9) and included reports on these results in the revised manuscript.

In the revised manuscript, we point out the change in immobilization strategy using biotinylated target RNA as part of ternary complexes analyzed in intramolecular FRET measurements: ... For intermolecular and intramolecular FRET measurements, we used a biotinylated target RNA for the immobilization of ternary complexes to ensure that only fully assembled ternary complexes are immobilized and included in these measurements. ...

Comment:

The section describing results in Fig. 5 is confusing and seems illogical at points. For example: "Since bulges and the hairpin are located in the 3'-region of the miRNA and therefore in the N-terminal lobe." This reasoning seems incongruent with the structure of pre-miR451 in Fig. 5E, which has no notable bulges (only a G:U wobble pair at the very end of the duplex). And, "Similar data were collected for the guide/target combination with two bulges in the central and supplementary region of the guide (Figure 5D)." The guide and target in Fig. 5D have so little complementarity after the seed that I do not believe it is possible to reliably predict the structure of any potential duplex that might form (especially while bound to Ago2).

Response:

We thank the reviewer for pointing this out to us. We fully agree that this description is confusing and we changed the respective paragraphs and pointed out to the reader that there is little to no complementarity in the central, supplementary and tail region, whereas there is a full match in the seed region.

Changes:

... we tested the miRNA451, which displays only seed complementarity to its target and little to no complementarity in the central, supplementary and tail region of the guide RNA...

... Since the hairpin and the non-paired region of the guide-target duplex are located in the 3'-region of the miRNA...

... Similar data were collected for the guide/target combination with complementarity only in the seed region...

... is disfavored in case of bulged or mismatched substrates...

... as well as the unpaired part of the target RNA demand space....

Minor comments:

Comment:

Line 163/Fig 2C: The 0.66 nm distance larger than that observed in crystal structures in ref 10 is intriguing. Might it be connected to the observation that the RNAs in these crystal structures were held in place by crystal lattice contacts? Indeed, in one structure (PDB ID: 6MFN, see Fig. 1b in ref 10), which lacked lattice contacts, the portion of RNA containing g14 was completely disordered. I am therefore wondering: does the E1 peak in Fig 2C correspond to a well-defined conformation that could be observed via structural methods (but has gone unobserved so far), or does the E1 peak arise from a family of conformations that resemble structures in ref 10 (in terms of protein conformation), but are dynamic (in terms of RNA conformation) and thus be represented by PDB ID: 6MFN? Can the data distinguish between these two possibilities?

Response:

We agree that this conformation might not have been observed in crystal structures because of the crystal lattice contacts. The conformation reflected by population E1 is probably not as defined in terms of RNA positions as the conformations in population E3 that represents the conformation in which the guide's 3'-end is anchored in the PAZ binding pocket. Also, a range of 3'-end released conformations, (e.g. the catalytically active form) is structurally well-defined, otherwise cleavage would not be possible. In case of population E1, we suppose that base pairing between guide and target occurs in the seed as well as the supplementary region. In case of an unpaired supplementary region, reassociation of the guide's 3'-end with the PAZ binding pocket would be the more likely scenario. The bulge in between those two double stranded regions allows for a little flexibility as the chain in between the two sticks of a nunchuck does. The 5'-end of the guide is fixed to the Mid domain and is therefore in a defined position. In conclusion, the flexibility is limited to the supplementary region. However, the fact that we observe a defined population indicates that there is only limited flexibility of the RNA.

Comment:

Statement on lines 97-98 should include citation to PMID: 26140367. It would also be reasonable to cite PMID: 26140592, as mouse and human Ago2 are nearly identical.

Response:

We thank the reviewer for indicating the missing citation. We added the paper from Jo and colleagues (2015) but decided against adding the other paper as we specifically talk about human RISC in this sentence.

Changes:

Citation added

Comment:

Line 128 "seed-matching target (g1 – g8)" should be "seed-matching target (g2 – g8)" as g1 is not part of the seed and does not physically match with the target. This minor error occurs multiple other places in the manuscript.

Response:

We changed this throughout the manuscript and in the respective Figure.

Changes:

...with a seed-matching target (g2 – g8) resulted in three FRET populations...

...extended supplementary region but with a central bulge (g2-8 + g13-19)...

Comment:

Lines 137-144. The wording here carries a sense of absoluteness that is inaccurate. For example, "...according to structural studies, the 3'-end of the guide is still anchored in the PAZ domain..." indicates crystal structures provide a full view of the RNA-protein complex, which of course is not the case as crystal structures only provide snap-shots of conformations that are low energy enough to enable crystallization. Similarly, "...base pairing within the seed region is sufficient to trigger the release of the 3'-end from the PAZ domain..." does not reflect the observation that E3 remains the most populated state in the presence of the seed-matched target RNA. Perhaps "...is sufficient to stimulate the release...." is more accurate?

Response:

We fully agree that crystal structures can only provide structural snapshots while the conformational landscape of these protein-RNA complexes might be more multifaceted. We therefore changed the wording to avoid misinterpretation.

Changes:

...However, the crystallized state of a hAgo2-guide RNA complex represents a conformation in which the 3'-end of the guide is still anchored in the PAZ domain even in a seed plus supplementary paired state...

...We conclude that base pairing within the seed region is sufficient to stimulate the release of the 3'-end from the PAZ domain...

Comment:

Fig. 2: why does the distribution of E3 broaden in panels C and D (compared to A and B)? (this is not a criticism, just curiosity).

Response:

The broadening of a conformation indicates that there is more variety in the distance distribution among the molecules measured. This is usually due to a higher degree of conformational flexibility between the two fluorophores and by inference the PAZ domain and the 3'-end of the guide. We furthermore propose that molecules can still adopt the E3 state but additionally sample another state with a minor change in distance in the complexes shown in panel C and D. Both of these conformations are hidden in the E3 populations shown in panel C and D, which ultimately leads to a broadening of the FRET distribution. In case of panel D, this is also corroborated by the analysis of dynamic molecules shown in Figure S7 panel F (i) that confirms that the molecules adopt two states with a small change in FRET efficiency represented by the FRET efficiency of 0.59 in panel D.

Changes:

Comment:

Line 195-196: "Although crystal structures indicate that a centrally paired catalytically active ternary complex is relatively unstable (ref 9)". This statement is inaccurate. Ref 9 only noted that even when central pairing is available, Ago2 molecules can maintain a stable conformation in which central pairing is avoided. This is not evidence that a centrally paired catalytically active ternary complex is unstable or even less stable.

Response:

We thank the reviewer for pointing out this incorrect wording. Following the reviewer's suggestion, we changed this statement.

Changes:

...Although crystal structures show that ternary complexes with full pairing potential are able to adopt a stable conformation without central base pairing⁹,...

Comment:

Line 368: "Our data therefore rationalize the observed differences in guide RNA lifetimes that arise from different 3'-end release probabilities of TDMD targets and targets with full base pairing potential." This statement needs a reference.

Response:

We thank the reviewer for indicating this and apologize for overlooking this missing citation. We added the appropriate citations.

Changes:

...Our data therefore rationalize the observed differences in guide RNA lifetimes that arise from different 3'-end release probabilities of TDMD targets and targets with full base pairing potential^{10,30}....

Comment:

Line 29-30 "which ultimately leads to hAgo2-catalyzed degradation of the mRNA or translational inhibition" Consider "hAgo2-mediated degradation" or "hAgo2-directed degradation", as catalytic cleavage only works with special targets/precursors or artificial guides, while likely the majority of transcripts targeted by Ago2 are catalytically degraded by other enzymes.

Response:

We apologize for the imprecise phrasing and changed the wording accordingly.

Changes:

...which ultimately leads to hAgo2-mediated degradation of the mRNA...

Comment:

Line 77-78: "Binding of TDMD targets is accompanied by the release of the guide's 3'-end exposing this end for modification...thereby possibly triggering the proteolysis of Ago." This could be worded more precisely, as ref 33 and 34 both propose trimming-tailing-independent degradation of Ago.

Response:

We agree with the reviewer that this wording is imprecise and thank the reviewer for pointing this out to us. We changed the wording accordingly.

Changes:

...by the release of the guide's 3'-end¹⁰ exposing this end for modification^{30,31}. Furthermore, binding of TDMD targets possibly triggers conformational changes that lead to ubiquitinylation of hAgo2 resulting in the proteolysis of Ago^{33,34}....

Comment:

Figure 2 Legend E is missing.

Response:

We added the missing figure legend.

Changes:

Figure legend complemented.

Reviewer 2

remarks to the author:

Willkomm and colleagues examined the conformational dynamics of hAgo2 using site-specifically labelled hAgo2 and single-molecule FRET. The results obtained by the authors confirm previous studies and reveal new conformational states. They observed a catalytically active conformation and also reported a conformational state that is unique for TDMD-inducing targets. Overall, this well-designed and well-executed study provides new and interesting insights into the conformational dynamics of hAgo2. Additionally, the manuscript is well written. Therefore, this reviewer enthusiastically recommends this manuscript for publication in Nature Communications.

Minor comments:

Comment:

They authors should describe how they determine how many Gaussians to fit to each histogram.

Response:

We generally used the minimum number of Gaussians yielding an R^2 higher than 0.97 for the overall fit, but also included a priori knowledge about the state distribution that we gained from dynamic

FRET traces (i. e. compare exemplary FRET traces, transition density plots and histograms covering dynamic molecules) as well as measurement with single Gaussian fits.

Changes:

We added a description of the fitting process to the methods section.

Comment:

While FRET is sensitive to nanometer distance changes, it cannot be and should not be used to estimate an absolute distance. For example, the description in line 160 ("The large corresponding distance of 6.6 nm suggests ...") should be moderated.

Response:

We intended to point out to the reader that the distance between the guide supplementary region and the PAZ domain is substantially increased in comparison to all other guide/target combinations. However, to avoid confusion, we changed the wording.

Changes:

... The distance in between the PAZ and the guide RNA supplementary region in case of TDMD targets is larger than in case of all other guide/target combinations tested. This suggests a relocation of the guide away...

Comment:

Description of Figure 2E is missing in the figure caption.

Response:

We added the missing figure legend.

Changes:

Figure legend complemented.

Reviewer 3

remarks to the author:

The authors utilize a site-specific protein labeling approach (that they have published before) to introduce FRET labels into human Argonaute-2 (hAgo2). They then collectively use the labelled hAgo2 variants (complemented with labeled guide RNAs) to explore the structural versatility of apo-hAgo2 and to study its dynamic transitions upon guide RNA binding and target recognition in various experimental scenarios. Whereas parts of this study, especially regarding seed-target pairing, constitute primarily supportive evidence for prior results gained by structural biology, the study also reveals new, exciting insights into target-directed microRNA degradation, pre-miR451 processing and the cleavage-competent hAgo2 conformation. Provided that my concerns are adequately addressed, I see this study

as an important addition to previous structural work on hAgo2 and therefore suitable for publication in Nature Communications.

Major comments:

Comment:

The authors define the FRET label at hAgo2 residue F23 as being part of the PAZ-N lobe of hAgo2. This statement needs further rationalization (and explanation), as residue 23 of hAgo2 directly packs against a helix (675-694) in the PIWI domain of hAgo2. I therefore posit that the label is dynamically part of the MID-PIWI lobe, not the PAZ-N lobe. Given this discrepancy in interpretation, I urge the authors to rationalize their labeling choice. In particular, the authors need to experimentally clarify why the F23 label dynamically represents the N-lobe of hAgo2. Should this not be the case, the authors need to re-interpret their data throughout the entire study.

Response:

We thank the reviewer for drawing our attention to this point. In the revised version of the manuscript, we explain in detail the choice of labelling positions.

Changes:

We added the following sentences: “The label at position F23 was chosen because of its central position in between the lobes of the hAgo2 protein. Crystal structures of hAgo2 reveal that the N-terminus of hAgo2 is packed against the PIWI domain^{6,7}. Hence, phenylalanine 23, although nominal part of the N-terminal domain, is associated with the PIWI domain.” and changed some text passages in the respective paragraphs accordingly.

Comment:

Figs. S3A and S3B do not convince me of the authors’ statements that all hAgo2 eGFP fusion proteins are catalytically equally active and that the experimental conditions prevent target cleavage from occurring. More convincing biochemical evidence in support of the authors’ statement is required here.

Response:

Indeed, cleavage activities of the tested hAgo2 variants apparently differ in our *in vitro* cleavage assays. Unfortunately, these *in vitro* validation experiments are not quantitative, and the noise range only allows for qualitative conclusions. The experiments are performed with immunoprecipitated hAgo2 on beads and thus any short hAgo2 fragment that is, to some extent generated from the inserted stop codon, will also bind to the beads but not contribute to the cleavage reactions. Since this is different for each construct, we can only conclude that the hAgo2 full length variants are generally active, but our assays do not allow for a comparison between the individual hAgo2 constructs unfortunately. Thus, we have toned-down our conclusions in the text and simply state that activity is observed for all constructs suggesting that they are functional.

Changes:

.. We verified that modified hAgo2 variants were still active in guide-mediated RNA cleavage (**Figure S4A**)...

Comment:

The observed significant kink in a TDMD-prone guide/target duplex is intriguing and an interesting result of this study. However, to generalize this finding with regards to the mechanism of TDMD, the authors should at least use one additional miRNA/target RNA pair for their experiments.

Response:

We thank the reviewer for this suggestion and performed additional experiments with alternative TDMD substrates. We choose a miRNA/target RNA pair that was already used in a study conducted by the MacRae lab that elucidated structural details of target-directed miRNA degradation (doi: 10.1016/j.molcel.2019.06.019). Using this miRNA/target pair (new Supplementary Figure 6), we detected highly comparable datasets that support our previously conducted experiments (Figure 2C and 2D) and thereby strengthen our mechanistic conclusion that there is a yet undescribed conformation for TDMD substrates.

Changes:

Added a new supplementary figure (Supplementary Figure 6). Added passages that describe the new data in the revised version of the manuscript: "... To ensure that the large observed distance in between the PAZ domain and the supplementary region of the guide is associated with the TDMD pairing characteristics, we also tested another miRNA/target RNA pair (**Figure S6**). Indeed, the results from these experiments are in support of this hypothesis. ..." In addition, please see the section on fully matched target RNA.

Comment:

For a fully matched guide/target complex (Fig. 2D) the authors find two distinct populations. The one with lower FRET efficiency (0.35) they interpret as the catalytically active conformation, without providing any experimental proof for this statement. As the authors refer to this low-FRET population throughout the manuscript as the "active" conformation, I suggest that the authors set out to provide experimental evidence that this conformation correlates with hAgo2 activity.

Response:

We agree with the reviewer that an experiment that clearly associates the low FRET population with catalytic activity would be the final proof for our hypothesis that this low FRET population represents the catalytically active conformation. Since this is not possible in our very challenging experimental setup, we toned down our statements and emphasize that the association of the low FRET population with the catalytically active conformation is still a hypothesis. Furthermore, we expanded the reasoning for this hypothesis based on existing literature.

Changes:

... Even a target with full pairing potential to the hAgo2-bound guide allows for a stable conformation that keeps the guide 3'-end attached and the central cleft closed, thereby avoiding central base pairing¹⁰. As suggested by Sheu-Gruttadauria et al.⁹, central base pairing as a pre-requisite for cleavage activity requires a substantial opening of the central cleft, which causes tension in the guide 3'-end and, hence, its release from the PAZ domain. Sheng et al.⁴⁴ analyzed the catalytic pre-cleavage state for *T. thermophilus* Ago, which also includes the guide 3'-end release from the PAZ domain accompanied by several other conformational changes that enable catalytic activity. Taken together, permanent release of the guide's 3'-end from the PAZ domain is a crucial step to the

formation of catalytically active ternary complexes. Hence, we suggest that the dominant low FRET population in this measurement ($E = 0.35$) reflects this catalytically active conformation....

... Although crystal structures show that ternary complexes with full pairing potential are able to adopt a stable conformation without central base pairing⁹, cleavage activity and our smFRET data disclose a frequent sampling of other states likely including the active state....

... we also expect a conformation, which might represent the catalytically active complex, which supposedly involves a re-location of the guide's supplementary region into the N-PIWI channel. The low FRET population (**Figures S5F and S8B**) likely represents this conformation. ...

...This probably includes the catalytically active conformation...

Minor comments:

Comment:

As I found the results of their study more exciting than the presented abstract, I encourage the authors to consider rewriting parts of the abstract to better reflect their results.

Response:

We thank the reviewer for the assessment of our study and rephrased the abstract.

Changes:

We added a new abstract.

Comment:

The cartoons symbolizing the respective experimental FRET setup are crucial to aid understanding. If possible, their relative size (with respect to the entire figure) should be increased to make subtle differences visible.

Response:

We agree with the reviewer that these cartoons are very important and thank the reviewer for pointing this out to us. To improve the readability of the manuscript, we increased the size of the cartoons.

Changes:

All figures with cartoons were edited.

Comment:

Fig. S2B: The authors refer to low-molecular weight protein products as "abortion products". What is meant by this? The doubly-labelled hAgo2 variants show significant impurities. Did the authors analyze the nature of these bands? Are protein yields sufficient to run Coomassie-stained SDS PAGE?

Response:

Bioorthogonal incorporation of unnatural amino acids frequently leads to abortion of translation at the incorporation site or after incorporation of the UAA. This gives rise to abortion products. These products are partially visible in the fluorescence scan, if the label is positioned within this abortion

product. Based on the fluorescence scan (S2B) and corresponding Western blot (S2C), we further characterized these abortion products and included them into the schematic depiction of the hAgo2 construct (S2A). Only in case of the double mutants, the abortion products are visible in significant amounts in the fluorescence scan. The single mutant abortion products do not carry a fluorescent label, because the translation stops prior to the incorporation site (Figure S2). The abortion products observed in the fluorescence scan of the double mutants only carry one fluorescence label (see schematic depiction shown in Figure S2A). Consequently, these molecules will not exhibit FRET. Furthermore, a significant portion of the protein - especially in the 5'-end binding region of the guide strand - is missing, which disables binding to guide and consequently target nucleic acids.

Changes:

We included a Western blot with a primary antibody directed against the N-terminus of hAgo2 into Supplementary Figure S2C to further characterize the abortion products. A schematic depiction of abortion products in Figure S2A completes this analysis.

Comment:

Fig.5: The authors refer to the miR-451-target interaction as possessing two bulges. As judged from the corresponding figure (Fig. 5D), I am unable to comprehend this statement.

Response:

We changed the wording in this paragraph in the revised version of the manuscript to improve the understanding of this passage.

Changes:

...which displays only seed complementarity to its target and little to no complementarity in the central, supplementary and tail region of the guide RNA...

Comment:

Figures are not consecutively arranged according to the flow of the manuscript, e.g. Figs. 5D and 5E are being referred to after Fig. 5B is. I urged the authors to correct this to aid the reader.

Response:

We thank the reviewer for pointing this out to us. We changed the order of the figure panels to follow the flow of the manuscript.

Changes:

We changed the order of the panels accordingly.

Comment:

How can the role of chaperones in hAgo2 loading and the authors' FRET data on apo-hAgo2 be integrated? I missed any comments on this potential connection, e.g. in the discussion. The authors may want to include thoughts in this direction in their manuscript.

Response:

We agree with the reviewer and added a paragraph in the discussion that emphasizes the role of chaperones in hAgo2 loading and discuss this in the light of our results.

Changes:

We added a paragraph in the discussion section.

REVIEWERS' COMMENTS

Reviewer #1 (Remarks to the Author):

The authors have addressed all of my previous concerns and criticisms with great care. Specifically, the authors did an impressive job quantifying and enriching apo-Ago2 molecules in their preparations, adding a great deal of confidence to their conclusions. I have nothing more to offer except to restate my gratitude for this excellent contribution to the field.

Reviewer #2 (Remarks to the Author):

In the first review, I was supportive of publishing this work but addressed a concern that FRET efficiency should not be used to estimate an absolute distance. The authors changed the wording for the example I gave them (previously line 160). However, in the revised manuscript, they still report distances based on the FRET values, e.g. lines 119 and 310. All the cases including these should be corrected before publication is considered.

Reviewer #3 (Remarks to the Author):

Overall, the authors addressed my concerns adequately and I do recommend publication of this work at this time.

An exception to this overall very positive assessment is major point #1 of my previous report. Here, the authors confirm my voiced concern by adding the following sentences to their manuscript: "The label at position F23 was chosen because of its central position in between the lobes of the hAgo2 protein. Crystal structures of hAgo2 reveal that the N-terminus of hAgo2 is packed against the PIWI domain. Hence, phenylalanine 23, although nominal part of the N-terminal domain, is associated with the PIWI domain." In essence, this statement confirms that the F23 label conformationally represents the PIWI domain, NOT the N-domain of hAgo2.

However, in contradicting their own statement, the authors write in lines 103-106: "We produced three hAgo2 variants to analyze intra-lobe rearrangements (hAgo2N/PAZ (fluorophores coupled to residue 23 and 291) and hAgo2N/Mid (residue 23 and 511)) and inter-lobe domain movements (hAgo2PAZ/Mid (residue 291 and 511)) (Figure 1A and S1B)." Here, measuring hAgo2N/PAZ data does not reveal *intra*-lobe rearrangements, but *inter*-lobe rearrangements. This is the case as the N-domain label at F23 conformationally represents the PIWI domain of hAgo2.

I therefore request the authors to change the wording of this paragraph to not confuse the reader. Moreover, the same inaccuracy in lobe vs. label definition applies to the paragraph in lines 303-321. Here the authors need to make sure to rightly distinguish intra- from inter-lobe rearrangements whenever utilizing hAgo2 labeled at F23.

Point-by-point response:

We thank the reviewers for the positive reception of the changes we added to our manuscript and the careful evaluation of the revised manuscript. In the current revised version of the manuscript, we addressed all concerns raised by the reviewers and changed the manuscript accordingly. Please find our point-by-point response below.

Reviewer 1

remarks to the author:

The authors have addressed all of my previous concerns and criticisms with great care. Specifically, the authors did an impressive job quantifying and enriching apo-Ago2 molecules in their preparations, adding a great deal of confidence to their conclusions. I have nothing more to offer except to restate my gratitude for this excellent contribution to the field.

Response:

We thank the reviewer for the very positive assessment of our revised manuscript.

Reviewer 2

remarks to the author:

In the first review, I was supportive of publishing this work but addressed a concern that FRET efficiency should not be used to estimate an absolute distance.

Comment:

The authors changed the wording for the example I gave them (previously line 160). However, in the revised manuscript, they still report distances based on the FRET values, e.g. lines 119 and 310. All the cases including these should be corrected before publication is considered.

Response:

We followed the request of the reviewer and changed our explanations accordingly.

Changes:

We removed all distance information from the text and the corresponding methods part and modified our explanations accordingly. The passages read as follows:

L. 123 – 128

... reflects a single conformational state of the guide nucleic acid within hAgo2. This correlates well with information derived from crystal structures of hAgo2-guide complexes. Crystal structures suggest a stable positioning of the guide within hAgo2 by (i) anchored 3'- and 5'-guide ends and (ii) interactions of hAgo2 with the guide backbone that lead to a pre-organized seed region and a stable positioning of the supplementary region⁶⁻⁸ (Figure 1C, ii and S1B).

However, crystal structures of ternary complexes comprising a target with TDMD characteristics do not reveal such a dramatic increase of the distance between the PAZ domain and the guide supplementary region¹⁰ compared to e.g. a seed-matched ternary complex (Figure S1B). Hence, our measurements indicate a higher degree of kinking.

Reviewer 3

remarks to the author:

Overall, the authors addressed my concerns adequately and I do recommend publication of this work at this time.

Comment:

An exception to this overall very positive assessment is major point #1 of my previous report. Here, the authors confirm my voiced concern by adding the following sentences to their manuscript: ‘The label at position F23 was chosen because of its central position in between the lobes of the hAgo2 protein. Crystal structures of hAgo2 reveal that the N-terminus of hAgo2 is packed against the PIWI domain. Hence, phenylalanine 23, although nominal part of the N-terminal domain, is associated with the PIWI domain.’ In essence, this statement confirms that the F23 label conformationally represents the PIWI domain, NOT the N-domain of hAgo2.

However, in contradicting their own statement, the authors write in lines 103-106: ‘We produced three hAgo2 variants to analyze intra-lobe rearrangements (hAgo2N/PAZ (fluorophores coupled to residue 23 and 291) and hAgo2N/Mid (residue 23 and 511)) and inter-lobe domain movements (hAgo2PAZ/Mid (residue 291 and 511)) (Figure 1A and S1B).’ Here, measuring hAgo2N/PAZ data does not reveal *intra*-lobe rearrangements, but *inter*-lobe rearrangements. This is the case as the N-domain label at F23 conformationally represents the PIWI domain of hAgo2.

I therefore request the authors to change the wording of this paragraph to not confuse the reader. Moreover, the same inaccuracy in lobe vs. label definition applies to the paragraph in lines 303-321. Here the authors need to make sure to rightly distinguish intra- from inter-lobe rearrangements whenever utilizing hAgo2 labeled at F23.

Response:

We agree with the reviewer that - although nominal part of the N-terminal domain - residue F23 is associated with the PIWI domain. Hence, as the reviewer pointed out, F23 can be considered part of the N-terminal or the C-terminal lobe depending on the definition making it difficult to use the term “intra-lobe” for the variants hAgo2-N/PAZ or hAgo2-N/Mid. However, as we now specified in the explanations, we chose F23 because it is a rather stable central residue in the center of the protein in between the lobes. In our measurements we aimed to analyze inter-lobe dynamics and employed residue H291 and T511 as labeling positions. We found extensive movements in the apo enzyme that decrease upon binding of guide and target nucleic acids. In a second step, we wanted to explore whether the whole lobe moves, or if there is movement along an intra-lobe axis, too. We consider these axes reaching from the PAZ or the Mid domain, respectively, to the center of the protein in between the lobes. Therefore, we chose the pivot point residue F23. We did not observe movements along this intra-lobe axes in the apo enzyme.

Changes:

We changed the respective paragraphs to emphasize our choice of the labeling position. The re-phrased passages read as follows:

L. 102 – 110

... We produced three doubly labeled hAgo2 variants. Two of these mutants were employed to analyze rearrangements of the Mid and the PAZ domain along the respective intra-lobe axis, using the N-terminal residue phenylalanine 23 as a second labeling position (hAgo2^{N/PAZ} (fluorophores coupled to residue 23 and 291) and hAgo2^{N/Mid} (residue 23 and 511)). We chose position 23 for labeling because of its central position in between the lobes of hAgo2. Crystal structures of hAgo2 reveal that this position at the N-terminus of hAgo2 is stably packed against the PIWI domain^{6,7}. Hence, phenylalanine 23, although nominal part of the N-terminal domain, is associated with the PIWI domain and stably positioned as a pivot point in between the lobes. The third mutant used was chosen to monitor inter-lobe domain movements (hAgo2^{PAZ/Mid} (residue 291 and 511)) (**Figure 1A** and **S1B**).

L. 309-314

To collect information about conformational changes along the intra-lobe axes, we examined conformations of the PAZ or the Mid domain in relation to residue F23, which marks a central pivot point between the two lobes of hAgo2. Because of its stable central position, we employed position 23 in combination with the labels in the PAZ or the Mid domain (hAgo2^{N/PAZ} and hAgo2^{N/Mid}) to gain information about conformational changes of these domains within the N- or the C-terminal lobe, respectively (**Figure 4** and **5**).

L. 389 – 390

Analysis of substrates with bulges in the supplementary region reveal a restriction of intra-lobe motions in the N-lobe.

L. 880 Legend Figure 6:

In addition, the axes in between the F23 and the Mid or the PAZ domain are mobile suggesting an engulfment of the bound RNA substrates